# Brain-specific Drp1 regulates postsynaptic endocytosis and dendrite formation independently of mitochondrial division

Kie Itoh[1], Daisuke Murata[1], Takashi Kato[1], Tatsuya Yamada[1], Yoichi Araki[2], Atsushi Saito[3], Yoshihiro Adachi[1], Atsushi Igarashi[1], Shuo Li[1], Mikhail Pletnikov[2,3], Richard L Huganir[2], Shigeki Watanabe[1,2], Atsushi Kamiya[3], Miho Iijima[1]*, Hiromi Sesaki[1]*

[1]Department of Cell Biology, Johns Hopkins University School of Medicine, Baltimore, United States; [2]Solomon H. Snyder Department of Neuroscience, Johns Hopkins University School of Medicine, Baltimore, United States; [3]Department of Psychiatry and Behavioral Sciences, Johns Hopkins University School of Medicine, Baltimore, United States

**Abstract** Dynamin-related protein 1 (Drp1) divides mitochondria as a mechano-chemical GTPase. However, the function of Drp1 beyond mitochondrial division is largely unknown. Multiple Drp1 isoforms are produced through mRNA splicing. One such isoform, Drp1$_{ABCD}$, contains all four alternative exons and is specifically expressed in the brain. Here, we studied the function of Drp1$_{ABCD}$ in mouse neurons in both culture and animal systems using isoform-specific knockdown by shRNA and isoform-specific knockout by CRISPR/Cas9. We found that the expression of Drp1$_{ABCD}$ is induced during postnatal brain development. Drp1$_{ABCD}$ is enriched in dendritic spines and regulates postsynaptic clathrin-mediated endocytosis by positioning the endocytic zone at the postsynaptic density, independently of mitochondrial division. Drp1$_{ABCD}$ loss promotes the formation of ectopic dendrites in neurons and enhanced sensorimotor gating behavior in mice. These data reveal that Drp1$_{ABCD}$ controls postsynaptic endocytosis, neuronal morphology and brain function.

DOI: https://doi.org/10.7554/eLife.44739.001

*For correspondence:
miijima@jhmi.edu (MI);
hsesaki@jhmi.edu (HS)

Competing interests: The authors declare that no competing interests exist.

## Introduction

The major function of Drp1, which is encoded by the *Dnm1l* gene, is to control mitochondrial division as a mechano-chemical GTPase (*Kameoka et al., 2018*; *Kraus and Ryan, 2017*; *Pernas and Scorrano, 2016*; *Prudent and McBride, 2017*; *Ramachandran, 2018*; *Tamura et al., 2011*; *van der Bliek et al., 2013*). During mitochondrial division, Drp1 is assembled into helical filaments around the surface of mitochondria. Through GTP hydrolysis and interactions with receptors, the Drp1 filaments change their conformation and constrict the mitochondrial membrane. Mitochondrial division is important for human health: hyper- or hypo-division caused by the mis-regulation of Drp1 has been linked to many neurological disorders, such as Alzheimer's, Parkinson's, and Huntington's diseases (*Cho et al., 2010*; *Itoh et al., 2013*; *Kandimalla and Reddy, 2016*; *Roy et al., 2015*; *Serasinghe and Chipuk, 2017*). Notably, human Drp1 mutations also lead to neurodevelopmental defects with post-neonatal lethality, developmental delay, late-onset neurological decline, or optic atrophy (*Fahrner et al., 2016*; *Gerber et al., 2017*; *Vanstone et al., 2016*; *Waterham et al., 2007*); however, our current understanding of Drp1's function outside of mitochondrial division is limited.

To study the function of Drp1, complete and tissue-specific knockout (KO) mice for Drp1 have been characterized. The loss of Drp1 results in mitochondrial elongation and enlargement due to

unopposed mitochondrial fusion in the absence of mitochondrial division in many cells (*Friedman and Nunnari, 2014*; *Kashatus, 2018*; *Widlansky and Hill, 2018*; *Youle and van der Bliek, 2012*). Complete loss causes embryonic lethality (*Ishihara et al., 2009*; *Wakabayashi et al., 2009*), whereas neuron-specific KO leads to a wide range of phenotypes, depending on the types of neurons and the timings when Drp1 is knocked out. For example, the loss of Drp1 in cerebellar Purkinje cells results in developmental defects when knocked out in embryos and progressive degeneration when knocked out in post-mitotic adult Purkinje cells (*Kageyama et al., 2012*; *Wakabayashi et al., 2009*). Similar to Purkinje cells, the loss of Drp1 induces massive death in dopaminergic neurons (*Berthet et al., 2014*). In contrast, hippocampal neurons are more resistant to the loss of Drp1; hippocampal neurons that lack Drp1 or express dominant negative Drp1, do not die but instead show deficits in bioenergetic and synaptic functions (*Divakaruni et al., 2018*; *Shields et al., 2015*). Similarly, Drp1-KO hypothalamic pro-opiomelanocortin neurons are also viable and show increased glucose and leptin sensing (*Santoro et al., 2017*).

Drp1 is encoded by a single gene and produces multiple isoforms through alternative splicing of mRNAs. There are four alternative exons in Drp1 in mice (termed A, B, C, and D) (*Figure 1A*). These alternative exons are located in either the GTPase domain (A and B) or the variable domain (C and D), which is mainly intrinsically disordered and contains regulatory phosphorylation sites (*Itoh et al., 2018*). All of the Drp1 isoforms are located at mitochondria and function in mitochondrial division (*Itoh et al., 2018*). Interestingly, a subset of these isoforms is also located at additional sites. For example, $Drp1_D$ and $Drp1_{BD}$ are associated with and regulate the dynamics of microtubules (*Itoh et al., 2018*; *Strack et al., 2013*). We recently identified a novel isoform of Drp1 (termed $Drp1_{ABCD}$) that is exclusively expressed in the brain (*Itoh et al., 2018*). $Drp1_{ABCD}$, which contains all of the alternative exons, is the only isoform that is associated with lysosomes, late endosomes, and the plasma membrane when this isoform is expressed in Drp1-KO mouse embryonic fibroblasts (MEFs) (*Itoh et al., 2018*). Analysis of transcripts and proteins showed that $Drp1_{ABCD}$ is expressed at low levels; $Drp1_{ABCD}$ constitutes less than 5% of all the Drp1 isoforms expressed in the brain (*Itoh et al., 2018*).

The unique localization of $Drp1_{ABCD}$ suggests that this brain-specific isoform may play a role in membrane trafficking in neurons; however, its function remains to be determined because of the lack of tools to specifically assess its function without affecting other isoforms. In this study, we have developed isoform-specific knockdown by shRNA and knockout by CRISPR/Cas9. Using these new approaches, we found that $Drp1_{ABCD}$ controls postsynaptic endocytosis and dendrite growth in neurons independently of mitochondrial division.

## Results and discussion

### $Drp1_{ABCD}$ expression is induced during postnatal brain development

Since $Drp1_{ABCD}$ is the only isoform that contains both the alternative exons A and B (*Figure 1A*), we raised antibodies that specifically recognize $Drp1_{ABCD}$ using the amino acid sequence that corresponds to the junction of exons A and B as an antigen (*Itoh et al., 2018*). The expression of $Drp1_{ABCD}$ was spatially controlled and specific to the brain (*Itoh et al., 2018*) (*Figure 1B*). In the brain, $Drp1_{ABCD}$ was ubiquitously expressed in multiple subregions, including the hippocampus, cortex, midbrain, striatum, and cerebellum (*Itoh et al., 2018*).

To test whether the expression of $Drp1_{ABCD}$ is temporally regulated in the brain, we performed Immunoblotting of whole brain and hippocampus tissues that were harvested from mice at the ages of P0, P8, 1 month, and 2 months. We found that the expression of $Drp1_{ABCD}$ is postnatally induced in both tissues later in neural development compared to that of postsynaptic density protein 95 (Psd-95), which is a synaptic protein required for glutamate receptor organization (*Figure 1C*). In contrast, anti-pan-Drp1 antibodies, which recognize all Drp1 isoforms, showed similar levels of Drp1 at different stages of postnatal brain development (*Figure 1C*). Consistent with the in vivo data, immunoblotting of hippocampal neurons cultured in vitro showed that the expression of $Drp1_{ABCD}$ gradually increases and reaches a plateau around 3 weeks (*Figure 1D*).

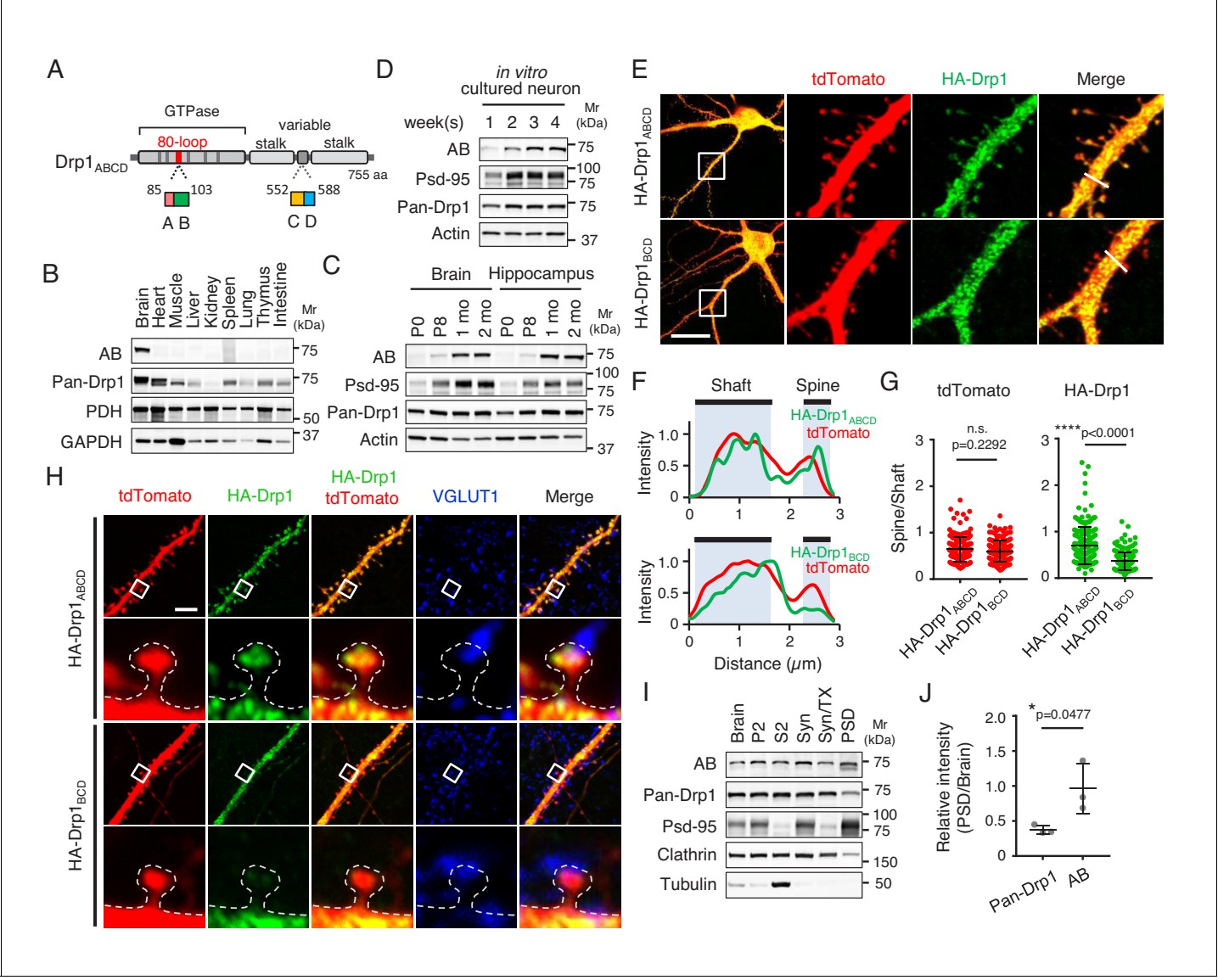

**Figure 1.** Drp1$_{ABCD}$ is induced during a postnatal period and enriched in postsynaptic terminals. (**A**) Domain architecture of Drp1$_{ABCD}$. Alternative exons A and B are present in the 80-loop inside the GTPase domain while alternative exons C and D are located in the variable domain. (**B**) Different mouse organs were analyzed by Immunoblotting using antibodies to Drp1$_{ABCD}$ (AB), pan-Drp1, the mitochondrial protein PDH, and GAPDH. 60 μg (AB and pan-Drp1) and 12.5 μg (PDH and GAPDH) of proteins were loaded per lane. (**C**) Whole brains and hippocampi were analyzed at the indicated ages by Immunoblotting with antibodies to Drp1$_{ABCD}$, postsynaptic density protein 95 (Psd-95), pan-Drp1, and actin. (**D**) Hippocampal neurons were cultured in vitro for 1, 2, 3 and 4 weeks and analyzed by immunoblotting. (**E**) Cultured hippocampal neurons were co-transfected at 3 weeks with plasmids carrying HA-Drp1$_{ABCD}$ or HA-Drp1$_{BCD}$, along with plasmids carrying a cytosolic marker, tdTomato. Three days after transfection, neurons were analyzed by immunofluorescence microscopy with antibodies to RFP (which recognizes tdTomato) and HA. Boxed regions are enlarged. Bar, 20 μm. (**F**) Intensity of tdTomato (red) and HA (green) signals in dendritic shafts and spines were quantified along the lines shown in *Figure 1E*. Intensity was normalized to the highest value. (**G**) Ratios of signal intensity in spines relative to those in dendritic shafts were analyzed for HA-Drp1$_{ABCD}$ and HA-Drp1$_{BCD}$. As a control, the tdTomato signal was used. Bars are mean ± SD (n = 176 spines in 10 neurons expressing HA-Drp1$_{ABCD}$ and 163 spines in 10 neurons expressing HA-Drp1$_{BCD}$). (**H**) Cultured hippocampal neurons were co-transfected at 3 weeks with plasmids carrying tdTomato and HA-Drp1$_{ABCD}$ or HA-Drp1$_{BCD}$ and subjected to immunofluorescence microscopy with antibodies to HA and vesicular glutamate transporter 1 (VGLUT1). Boxed regions are enlarged. Bar, 5 μm. (**I**) Postsynaptic density fractions were isolated from the whole brains of wild-type mice and analyzed by Immunoblotting. Brain, whole brain; P2, membrane fraction; S2, cytosolic fraction; Syn, total synaptosomal fraction; Syn/Tx, Triton-soluble synaptosomal fraction; PSD, postsynaptic density fraction. (**J**) Band intensity of total Drp1 (pan-Drp1) and Drp1$_{ABCD}$ (AB) in the postsynaptic density fraction was quantified relative to the whole brain. Bars are mean ± SD (n = 3). Statistical analysis was performed using Mann–Whitney $U$ test (**G**) and Student's $t$-test (**J**). n.s., not significant.

DOI: https://doi.org/10.7554/eLife.44739.002

*Figure 1 continued on next page*

*Figure 1 continued*

The following source data is available for figure 1:

**Source data 1.** Drp1$_{ABCD}$ is enriched in postsynaptic terminals.
DOI: https://doi.org/10.7554/eLife.44739.003

## Drp1$_{ABCD}$ is enriched in postsynaptic terminals

To examine the subcellular localization of Drp1$_{ABCD}$ in neurons, we expressed HA-Drp1$_{ABCD}$ along with a cytosolic marker, tdTomato, in cultured hippocampal neurons. For a comparison, we tested HA-Drp1$_{BCD}$, the most abundant brain Drp1 isoform (*Itoh et al., 2018*). In the soma, both HA-Drp1$_{ABCD}$ and HA-Drp1$_{BCD}$ appeared to be uniformly distributed (*Figure 1E*). We did not observe a clear association between these HA-tagged Drp1 isoforms and mitochondria, lysosomes, or the plasma membrane, likely due to their overexpression and therefore high levels in the cytoplasm. Interestingly, however, we found that HA-Drp1$_{ABCD}$ is enriched in postsynaptic regions, compared to HA-Drp1$_{BCD}$ (*Figure 1E*). Line scanning analysis of their fluorescence showed a significant increase in the signal ratio of HA-Drp1$_{ABCD}$ (spine vs dendritic shaft), compared to HA-Drp1$_{BCD}$ (*Figure 1F and G*). Analysis of HA-Drp1$_{ABCD}$ and HA-Drp1$_{BCD}$ in synapses at high magnification suggested its preferential localization of HA-Drp1$_{ABCD}$ around the postsynaptic density, which is in a close apposition to the pre-synaptic marker vesicular glutamate transporter 1 (*Figure 1H*).

To test the localization of endogenous Drp1$_{ABCD}$ at the postsynaptic density, we biochemically obtained postsynaptic density fractions from the brains of mice (*Araki et al., 2015*) since anti-Drp1$_{ABCD}$ antibodies do not work in immunofluorescence of the endogenous protein. Consistent with the immunofluorescence data, increased levels of Drp1$_{ABCD}$ were co-fractionated with the postsynaptic density, compared to total Drp1 detected by pan-Drp1 antibodies (*Figure 1I and J*).

## The loss of Drp1$_{ABCD}$ inhibits endocytosis at postsynaptic terminals

To examine the function of Drp1$_{ABCD}$ at the postsynaptic density, we homozygously deleted exon A using the CRISPR/Cas9 genome editing system (termed Drp1$_{exonA}$-KO mice) since Drp1$_{ABCD}$ is the only isoform that contains this exon (*Itoh et al., 2018*) (*Figure 2A*). We confirmed the lack of Drp1$_{ABCD}$ proteins in Drp1$_{exonA}$-KO mice using Immunoblotting (*Figure 2B*). Consistent with a low expression level of Drp1$_{ABCD}$ (compared to that of other isoforms, such as Drp1$_{BCD}$) (*Itoh et al., 2018*), we found no gross changes in the total amount of Drp1 in Immunoblotting using anti-pan-Drp1 antibodies (*Figure 2B*). Drp1$_{exonA}$-KO mice were born at an expected Mendelian ratio with normal weights of the body and brain (*Figure 2C–E*). H and E staining of sagittal brain sections showed that the histology of the cerebellum appears to be normal in Drp1$_{exonA}$-KO mice (*Figure 2F*). DAPI staining also showed similar nuclear patterns of neurons and the thickness of the CA1 layer in the hippocampus in control and Drp1$_{exonA}$-KO mice (*Figure 2G*). These data suggest that the loss of Drp1$_{ABCD}$ does not change the overall structure of the brain.

We isolated hippocampal neurons from E18.5 mouse embryos and cultured them in vitro for 3 weeks. We then examined synapses by transmission electron microscopy. In both control and Drp1$_{exonA}$-KO neurons, we observed matured synaptic contacts (*Figure 2H*). However, remarkably, the number of clathrin-coated pits (CCPs) was significantly different in these neurons—Drp1$_{exonA}$-KO neurons showed more CCPs in postsynaptic terminals compared with control neurons (*Figure 2H and I*). In contrast, the number of CCPs in presynaptic terminals was indistinguishable (*Figure 2H and J*). We then divided the morphologies of CCPs in postsynaptic terminals into three categories: shallow, U-shaped and omega-shaped pits. We found an increased frequency of shallow and U-shaped CCPs, which likely represent early stages during endocytosis, in Drp1$_{exonA}$-KO neurons (*Figure 2K and L*). The frequencies of omega-shaped CCPs were similar in control and Drp1$_{exonA}$-KO neurons (*Figure 2K and L*).

The observed increase in the number of CCPs could be explained by either activation or inhibition of endocytosis. First, the rate of clathrin-mediated endocytosis may be enhanced in Drp1$_{exonA}$-KO neurons and therefore shallow and U-shaped CCPs were observed at a higher frequency. Alternatively, clathrin-mediated endocytosis may be slowed at early stages after the initiation of endocytosis perhaps in Drp1$_{exonA}$-KO neurons and thereby the intermediates were accumulated. To distinguish between these two possibilities, we treated control and Drp1$_{exonA}$-KO neurons with dynasore, a

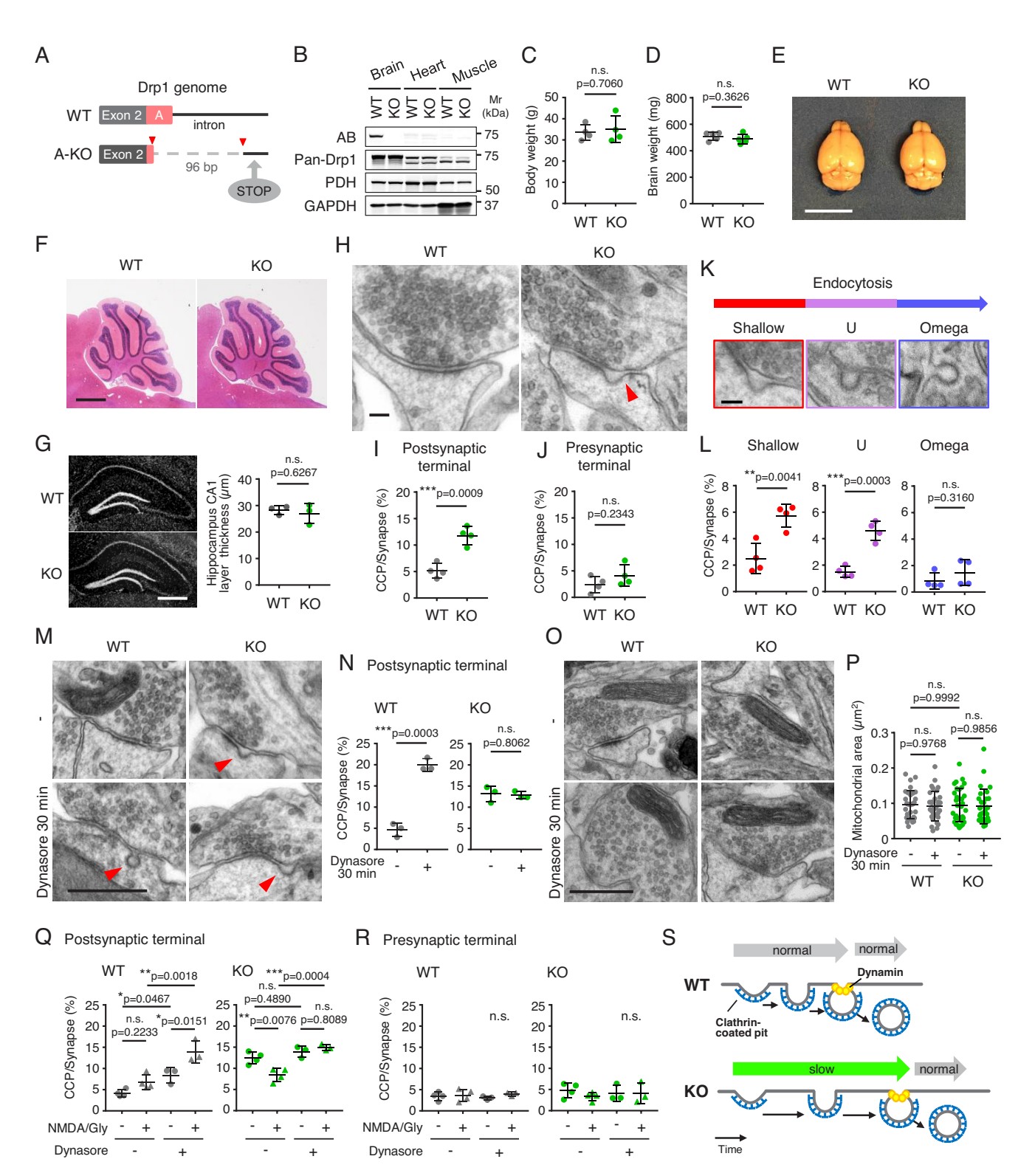

**Figure 2.** The loss of Drp1$_{ABCD}$ blocks postsynaptic endocytosis. (**A**) Two guide RNAs were used to cut the genome at two positions (red arrowheads) to remove the majority of exon A and part of the following intron using CRISPR/Cas9. This deletion introduced a stop codon 20 residues downstream from the deletion site (STOP). (**B**) The indicated tissues were harvested from control and Drp1$_{exonA}$-KO mice and analyzed by immunoblotting using antibodies to Drp1$_{ABCD}$ (AB), pan-Drp1, the mitochondrial protein PDH, and GAPDH. (**C and D**) Weights of the whole body (**C**) and brain (**D**) were
*Figure 2 continued on next page*

*Figure 2 continued*

measured. Bars are mean ± SD (n = 4 in C and 5 in D). (**E**) Images of the whole brain. Bar, 1 cm. (**F**) H and E staining of cerebella of control and Drp1$_{exonA}$-KO mice. Sagittal sections were cut in the midline. Bar, 1 mm. (**G**) Frozen sections of the hippocampus in control and Drp1$_{exonA}$-KO mice were stained with DAPI. Bar, 0.5 mm. The thickness of the CA1 layer was measured. Bars are mean ± SD (n = 3). (**H**) Control and Drp1$_{exonA}$-KO hippocampal neurons were cultured for 3 weeks and subjected to transmission electron microscopy. An arrowhead indicates a clathrin-coated pit (CCP) at a postsynaptic terminal. Bar, 100 nm. (**I and J**) Quantification of the number of CCPs at postsynaptic and presynaptic terminals. Bars are mean ± SD (n = 4 experiments, in which 167, 196, 172, 191 control and 158, 161, 169, 221 Drp1$_{exonA}$-KO synapses were analyzed). (**K and L**) The numbers of CCPs with three different morphologies (shallow, U-shaped, and Omega-shaped) were measured. Bar, 100 nm. (**M–P**) Control and Drp1$_{exonA}$-KO hippocampal neurons were treated with 80 μM of dynasore for 30 min and analyzed by electron microscopy (**M and O**). Bar, 500 nm. The number of CCPs (**N**) and the size of mitochondria (**P**) were determined. Bars are mean ± SD (n = 159, 182, 172 -/control, 152, 163, 143 +/control, 176, 163, 129 -/KO, and 162, 146, 145 +/KO synapses) (**N**) and (n = 30–32 mitochondria analyzed in each group) (**P**). (**Q and R**) Chemical long-term depression (NMDA/Gly) was induced by NMDA for 3 min in the presence or absence dynasore (80 μM). Neurons were then fixed, and CCPs at postsynaptic and presynaptic terminals were analyzed by electron microscopy. Bars are mean ± SD (n = 3–4 experiments. In each experiment, more than 100 synapses were analyzed). Statistical analysis was performed using Student's *t*-test (**C, D, G, I, J, L and N**) and One-way ANOVA with post-hoc Tukey (**P, Q and R**). (**S**) Summary of the data.

DOI: https://doi.org/10.7554/eLife.44739.004

The following source data and figure supplements are available for figure 2:

**Source data 1.** The loss of Drp1$_{ABCD}$ blocks postsynaptic endocytosis.

DOI: https://doi.org/10.7554/eLife.44739.007

**Figure supplement 1.** Dynasore does not affect mitochondrial morphology in cells.

DOI: https://doi.org/10.7554/eLife.44739.005

**Figure supplement 1—source data 1.** Data of mitochondrial length for B and C.

DOI: https://doi.org/10.7554/eLife.44739.006

dynamin inhibitor that blocks the final step of endocytosis (*Macia et al., 2006*), for 30 min prior to chemical fixation for electron microscopy. As expected, dynasore significantly increased the number of CCPs at postsynaptic terminals in control neurons (*Figure 2M and N*). In contrast, when we treated Drp1$_{exonA}$-KO neurons with dynasore, we found no increase in the number of CCPs (*Figure 2M and N*). Thus, it is likely that the rate of clathrin-mediated endocytosis is decreased in Drp1$_{exonA}$-KO neurons. The accumulation of shallow and U-shaped CCPs, but not omega-shaped ones, suggest that Drp1$_{ABCD}$ may function at an early step upstream of the constriction and severing of the neck of coated pits that is mediated by dynamin (*Figure 2S*). We confirmed that dynasore did not inhibit Drp1 by examining mitochondrial morphology in neurons and mouse embryonic fibroblasts using electron microscopy and immunofluorescence microscopy with antibodies to a mitochondrial protein (pyruvate dehydrogenase, PDH)(*Figure 2O and P*; *Figure 2—figure supplement 1*).

To further examine the consequence of Drp1$_{ABCD}$ loss in clathrin-mediated endocytosis, we stimulated WT and KO neurons with N-methyl-D-aspartic acid (NMDA) for a short period of time (3 min) and analyzed the number of CCPs at the postsypatic terminal. When we stimulated neurons in the presence of dynasore, the number of postsynaptic clathrin-coated pits increased in control neurons. This is due to stimulation of endocytosis by NMDA and inhibition of its completion by dynasore. In contrast, the number of CCPs remained unchanged in Drp1$_{exonA}$-KO neurons (*Figure 2Q*). These phenotypes of Drp1$_{ABCD}$ loss were only observed at the postsynaptic region and not the presynaptic region (*Figure 2R*). These data further support the notion that postsynaptic clathrin-mediated endocytosis is slow in Drp1$_{exonA}$-KO neurons even when stimulated by NMDA (*Figure 2S*). Interestingly, when stimulated by NMDA in the absence of dynasore, the number of CCPs was decreased in Drp1$_{exonA}$-KO neurons (*Figure 2Q*). It appears that NMDA induces internalization of some endocytic vesicles in Drp1$_{exonA}$-KO neurons. We suggest that Drp1$_{exonA}$-KO neurons have slow kinetics of endocytosis but do not completely block it (*Figure 2S*).

To understand how Drp1$_{ABCD}$ loss results in changes in CCPs, we tested postsynaptic positioning of the endocytic zone using mCherry-clathrin light chain (mCherry-CLC) and Psd-95-Fibronectin intrabodies (*Gross et al., 2013*; *Lu et al., 2007*). As previously reported (*Lu et al., 2007*), the majority of mCherry-CLC signals are localized next to Psd-95 signals in control neurons (*Figure 3A and B*). In contrast, we found a higher frequency of dissociation of mCherry-CLC signals from Psd-95 signals in Drp1$_{exonA}$-KO neurons (*Figure 3A and B*). We speculate that decreased levels of clathrin in the synapses in Drp1$_{exonA}$-KO neurons slow the progression of endocytosis. In these synapses, the

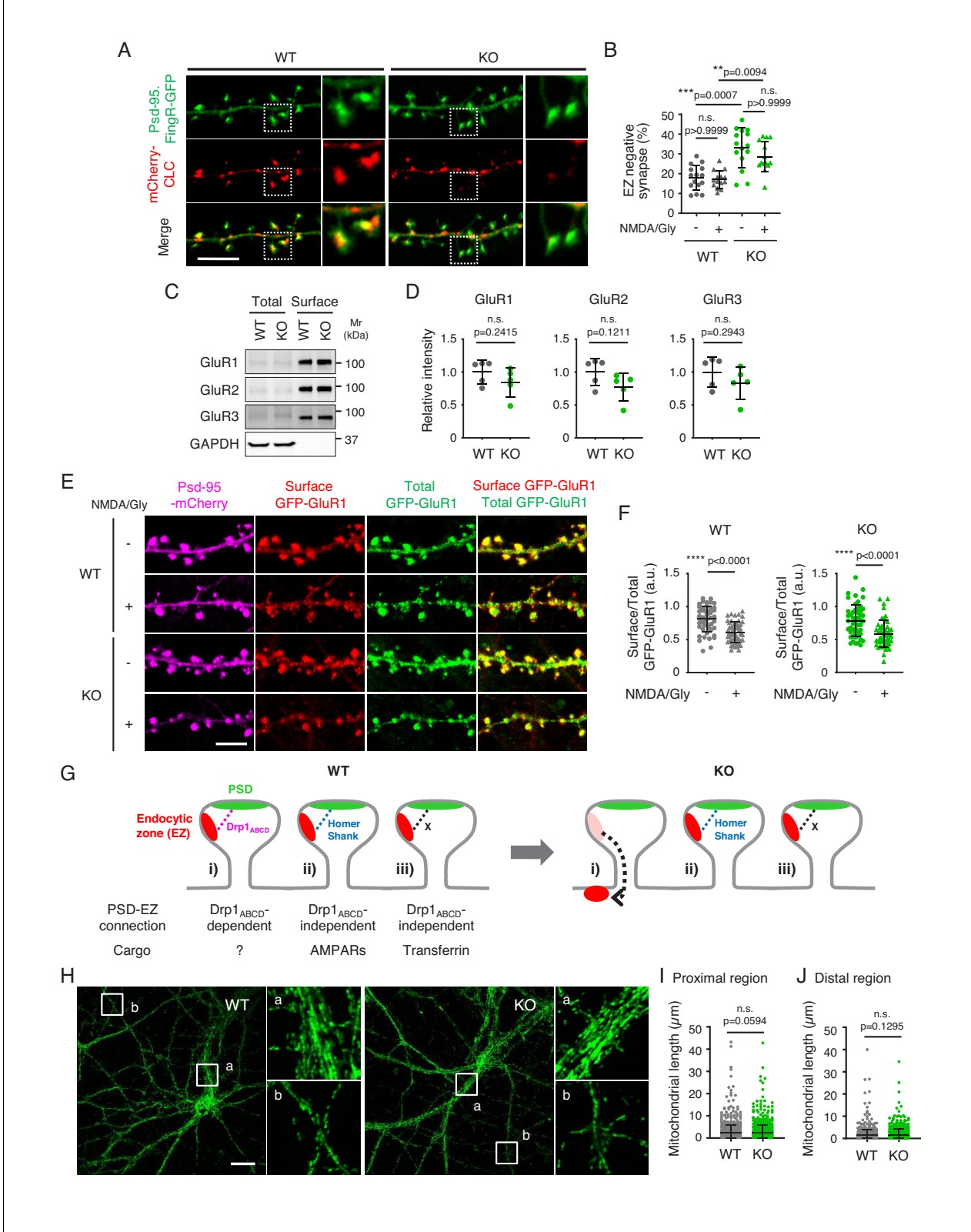

**Figure 3.** The endocytic zone is mislocalized in Drp1$_{exonA}$-KO neurons. (**A and B**) Hippocampal neurons were cultured and transfected with plasmids expressing Psd-95.FingR-GFP and mCherry-CLC. Two days after transfection, neurons were subjected to chemical LTD stimulation (NMDA/Gly), fixed and analyzed by laser scanning confocal microscopy. Bar, 5 μm. The number of Psd-95.FingR-GFP signals that are not associated with mCherry-CLC was scored. Bars are mean ± SD (n = 14–15 neurons were analyzed in each group). (**C**) Cell surface proteins were biotinylated with sulfo-NHS-SS-biotin

*Figure 3 continued on next page*

*Figure 3 continued*

in cultured control and Drp1$_{exonA}$-KO hippocampal neurons. The neurons were lysed and incubated with NeutrAvidin agarose. Total cell lysates and precipitated proteins (Surface) were separated by immunoblotting to antibodies to GluR1, GluR2, GluR3 and GAPDH. (D) Band intensity was determined. Bars are mean ± SD (n = 5). (E) Cultured neurons were co-transfected with plasmids carrying Psd-95-mCherry and GFP-GluR1. Two days after transfection, the neurons were treated with chemical LTD (NMDA/Gly) and subjected to immunofluorescence microscopy with anti-GFP antibodies without permeabilization of the plasma membrane. Images were acquired using identical settings. (F) The relative intensity of the signal from the anti-GFP antibodies (surface GFP-GluR1) compared with the GFP signal (total GFP-GluR1) was determined. Bars are mean ± SD (n = 50). (G) Model for the function of Drp1$_{ABCD}$ in the postsynaptic terminal. (H–J) Control and Drp1$_{exonA}$-KO hippocampal neurons were subjected to immunofluorescence microscopy with antibodies against the mitochondrial protein PDH. Boxed regions are enlarged: a, proximal dendritic regions and b, distal dendritic regions. Bar, 20 μm. The length of mitochondria was determined in proximal (I) and distal dendritic regions (J). Bars are mean ± SD (n = 10 neurons analyzed in each group. 70–120 mitochondria measured in each neuron). Statistical analysis was performed using Kruskal-Wallis test with Dunn's multiple comparisons test (B), Mann–Whitney *U* test (I) and Student's *t*-test (D, F and J).

DOI: https://doi.org/10.7554/eLife.44739.008

The following source data and figure supplements are available for figure 3:

**Source data 1.** The endocytic zone is mislocalized in Drp1$_{exonA}$-KO neurons.

DOI: https://doi.org/10.7554/eLife.44739.011

**Figure supplement 1.** Analysis of transferrin uptake.

DOI: https://doi.org/10.7554/eLife.44739.009

**Figure supplement 1—source data 1.** Data of transferrin uptake for A and B.

DOI: https://doi.org/10.7554/eLife.44739.010

formation of CCPs is initiated; however, the maturation of CCPs is likely decreased due to the limited availability of clathrin molecules. As a result, CCPs accumulate during relatively early stages of endocytosis (e.g., shallow and U-shaped CCPs) (*Figure 2L*). These data suggest that Drp1$_{ABCD}$, unlike dynamin, does not play a role in the scission of the neck of coated pits.

The extent of the dissociation of the postsynaptic density from the endocytic zone in Drp1$_{exonA}$-KO synapses is similar to that reported for the disruption of Homer, an adaptor protein that connects the postsynaptic density and endocytic zone (*Lu et al., 2007*). Like Homer defective neurons, the uptake of FITC-transferrin was not affected in Drp1$_{exonA}$-KO neurons (*Figure 3—figure supplement 1*). In contrast to the Homer pathway, however, we found that AMPA receptors, such as GluR1, GluR2 and GluR3, are normally expressed on the plasma membrane of Drp1$_{exonA}$-KO neurons in surface biotinylation experiments (*Figure 3C and D*). Furthermore, endocytosis of GFP-GluR1 in response to NMDA stimulation was not perturbed in Drp1$_{exonA}$-KO neurons (*Figure 3E and F*). These data suggest that the Drp1$_{ABCD}$ pathway has cargos that differ from those of the Homer pathway (*Figure 3G*).

Using immunofluorescence microscopy, we observed no gross changes in the morphology of mitochondria in proximal and distal regions along with dendrites in Drp1$_{exonA}$-KO neurons (*Figure 3H–J*). Therefore, inhibition of endocytosis does not appear to be the result of defects in mitochondrial morphology. It is likely that other Drp1 isoforms, such as Drp1$_{BCD}$ and Drp1$_{CD}$, which together constitute the majority of Drp1 isoforms in the brain (*Itoh et al., 2018*), mainly control mitochondrial division and morphology.

A previous study reported that Drp1 regulates endocytosis for synaptic vesicle recycling at presynaptic terminals in hippocampal neurons through interactions with a Drp1 receptor protein, Mff (*Li et al., 2013*). Since we found endocytic defects only at postsynaptic terminals in Drp1$_{exonA}$-KO neurons, distinct Drp1 isoforms may function separately in endocytosis at pre- and postsynaptic terminals.

## The loss of Drp1$_{ABCD}$ induces the extension of ectopic dendrites in cultured neurons

Intriguingly, during analysis of the morphology of cultured hippocampal neurons, we noticed that Drp1$_{exonA}$-KO neurons significantly increased the number of primary dendrites with dendritic spines (e.g., dendrites that directly emerged from the soma), compared to control neurons (*Figure 4A and B*). The number of axons that lack spines remained unchanged (one axon per neuron). The effect of Drp1$_{ABCD}$ loss was specific to the number of primary dendrites. We observed no significant

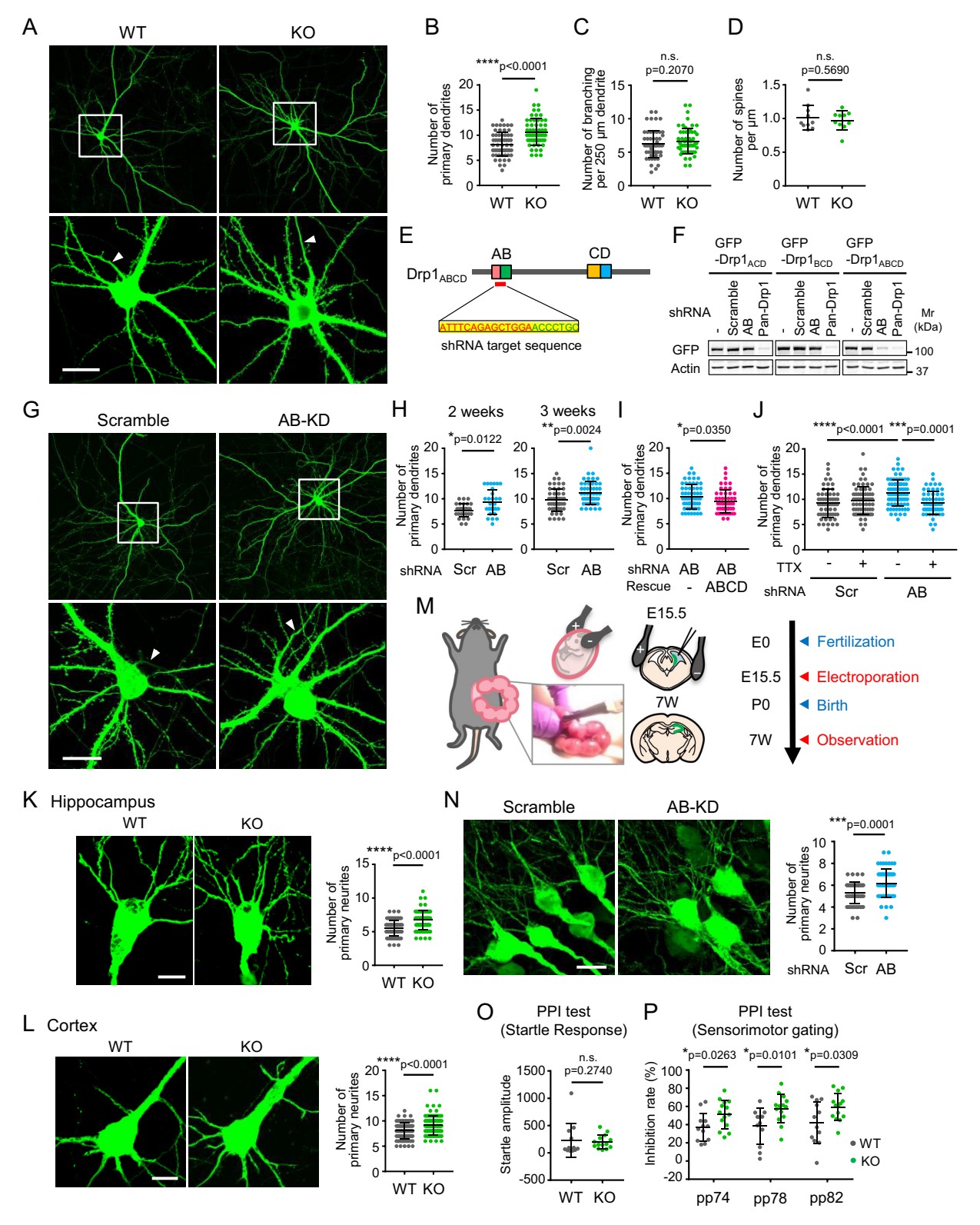

**Figure 4.** The loss of Drp1$_{ABCD}$ increases dendrite growth and sensorimotor gating. (**A**) Control and Drp1$_{exonA}$-KO hippocampal neurons were cultured and transfected with plasmids expressing GFP at 3 weeks. Boxed regions are enlarged. Arrowheads indicate axons that lack dendritic spines. Bar, 20 µm. (**B and C**) The numbers of primary dendrites (**B**) and dendritic branches (**C**) were quantified. Bars are mean ± SD (n = 60 control and 59 KO neurons). (**D**) The number of spines was quantified (n = 10 control and 10 KO neurons). (**E**) The DNA sequence that is targeted to knock down Drp1$_{ABCD}$

*Figure 4 continued on next page*

*Figure 4 continued*

is shown. (F) HEK293 cells were co-transfected with plasmids carrying the indicated GFP-Drp1 and shRNAs. Whole-cell extracts were analyzed by Immunoblotting using the indicated antibodies. (G) Mouse hippocampal neurons were cultured for 2 or 3 weeks and transfected with plasmids expressing the indicated shRNAs and GFP as a cytosolic marker. Images of 3 week cultured neurons are presented. Boxed regions are enlarged. Arrowheads indicate axons that lack dendritic spines. Bar, 20 μm. (H) The number of primary dendrites were quantified. Bars are mean ± SD (n = 29–30 neurons at 2 weeks and 50 neurons at 3 weeks). (I) Cultured neurons were transfected at 3 weeks with the plasmid expressing AB-targeted shRNA and GFP along with another plasmid carrying shRNA-resistant $Drp1_{ABCD}$. The number of primary dendrites was quantified. Bars are mean ± SD (n = 60 neurons for empty plasmid and 52 for $Drp1_{ABCD}$). (J) Cultured hippocampal neurons were transfected with the indicated shRNA plasmids that co-express GFP in the presence or absence of 2 μM tetrodotoxin (TTX). The number of primary dendrites was quantified (n = 60 for -TTX/scramble, 75 for +TTX/scramble, 79 for -TTX/AB and 57 for +TTX/AB). (K and L) Control and $Drp1_{exonA}$-KO mice were crossed with a mouse line expressing cytosolic GFP from the neuron-specific Thy1 promoter. We analyzed the number of neurites in the hippocampus (K) and cortex (L) at the age of 3–4 months. Bars are mean ± SD (n = 90 neurons in three mice for each genotype). Bar, 10 μm. (M) Plasmids carrying the indicated shRNAs were introduced into the hippocampi of E15.5 mouse embryos, along with plasmids carrying GFP, by electroporation in utero. (N) Hippocampi were analyzed at an age of 7 weeks using laser confocal microscopy of frozen brain sections. Bar, 20 μm. The number of neurites that directly emerge from the soma was quantified. Bars are mean ± SD (n = 51 neurons for scramble and 56 for AB-targeted). (O and P) Startle response (O) and PPI tests (P). Bars are mean ± SD (n = 12 control and 14 KO mice). Statistical analysis was performed using Student's *t*-test (**B, C, D. H-3 weeks, I, K, N and P**), Mann–Whitney *U* test (**H-2 weeks, L and O**) and One-way ANOVA with post-hoc Tukey (**J**).

DOI: https://doi.org/10.7554/eLife.44739.012

The following source data and figure supplements are available for figure 4:

**Source data 1.** The loss of $Drp1_{ABCD}$ increases dendrite growth and sensorimotor gating.
DOI: https://doi.org/10.7554/eLife.44739.018

**Figure supplement 1.** The number of axons is not affected by $Drp1_{ABCD}$ knockdown.
DOI: https://doi.org/10.7554/eLife.44739.013

**Figure supplement 2.** The expression of GFP from the Thy1 promoter.
DOI: https://doi.org/10.7554/eLife.44739.014

**Figure supplement 3.** Behavior analysis of $Drp1_{exonA}$-KO mice.
DOI: https://doi.org/10.7554/eLife.44739.015

**Figure supplement 3—source data 1.** Data of behavior tests.
DOI: https://doi.org/10.7554/eLife.44739.017

difference in the number of dendritic branches between control and $Drp1_{exonA}$-KO neurons (*Figure 4C*) or the density of dendritic spines (*Figure 4D*).

To further test whether $Drp1_{ABCD}$ controls dendrite formation in neurons in a cell-autonomous fashion, we specifically knocked down $Drp1_{ABCD}$ in cultured hippocampal neurons using shRNAs. To target $Drp1_{ABCD}$, we used an mRNA sequence that corresponds to the junction between exon A and exon B, which is unique to $Drp1_{ABCD}$ (*Figure 4E*). First, the specificity of this knockdown construct was confirmed. We individually expressed each of GFP-$Drp1_{ACD}$, GFP-$Drp1_{BCD}$, and GFP-$Drp1_{ABCD}$ in separate HEK293 cells. We found that AB-targeted shRNA specifically knocked down GFP-$Drp1_{ABCD}$, but not GFP-$Drp1_{ACD}$ or GFP-$Drp1_{BCD}$ (*Figure 4F*, AB shRNA). As a negative control, scramble shRNA was used (*Figure 4F*, Scramble). As a positive control, we targeted an mRNA sequence that is common in all Drp1 isoforms (*Figure 4F*, Pan-Drp1).

Supporting the data from the above experiments using $Drp1_{exonA}$-KO neurons, AB-targeted shRNA significantly increased the number of primary dendrites in cultured neurons at both 2 and 3 weeks compared to scramble shRNA (*Figure 4G and H*). Ectopic dendrites extended within a short period of time (3 days) after knockdown of $Drp1_{ABCD}$ in mature neurons with developed dendrites. The number of axons did not change (one axon per neuron) as assessed by immunofluorescence microscopy with anti-MAP2 antibodies, which label dendrites but not axons (*Figure 4—figure supplement 1*). To confirm that the induction of dendrite formation results from the knockdown of $Drp1_{ABCD}$, we co-expressed plasmids carrying a knockdown-resistant form of $Drp1_{ABCD}$ along with AB-targeted shRNAs. The $Drp1_{ABCD}$ plasmid, but not the empty plasmid, significantly rescued the effect of AB-targeted shRNAs (*Figure 4I*). These data further support the notion that $Drp1_{ABCD}$ is important for controlling the number of primary dendrites in neurons.

Dendrite growth is regulated by neuronal activity-dependent and -independent mechanisms (*Wong and Ghosh, 2002*). To understand the mechanism underlying the ectopic dendrite formation in AB-targeted neurons, we treated hippocampal neurons during knockdown with tetrodotoxin, a sodium channel inhibitor that blocks action potentials. We found that tetrodotoxin significantly

blocked the effect of Drp1$_{ABCD}$ knockdown on ectopic dendrite formation, but did not affect the number of dendrites in control neurons (*Figure 4J*). These data suggest that the formation of primary dendrites induced by Drp1$_{ABCD}$ depletion requires neuronal activity.

## Loss of Drp1$_{ABCD}$ induces the formation of ectopic primary dendrites in vivo

To test the function of Drp1$_{ABCD}$ in the morphology of neurons in vivo, we analyzed the morphology of neurons in Drp1$_{exonA}$-KO mice. To achieve this goal, it was critical to sparsely label individual neurons because the density of neurons is too high to faithfully visualize the morphology of each neuron if all of the neurons are labeled. We crossed Drp1$_{exonA}$-KO mice with a mouse line that expresses a cytosolic GFP in a small number of neurons driven by the neuron-specific Thy1 promoter (*Feng et al., 2000*) (*Figure 4—figure supplement 2*). We counted the number of neurites using z stacks of laser confocal microscopy of frozen brain sections. We found a significant increase in the number of neurites in the CA1 and CA2 layers in the dorsal hippocampus (*Figure 4K*) consistent with the data from the in vitro experiments. The effect of Drp1$_{ABCD}$ loss on primary dendrites was also evident in the cortex (*Figure 4L*).

To further test the effect of Drp1$_{ABCD}$ knockdown during brain development in vivo, we performed in utero electroporation of shRNAs. We injected plasmids carrying scramble or AB-targeted shRNA, along with plasmids carrying cytosolic GFP, into the lateral ventricles of E15.5 embryos in timed pregnant mice using a glass micropipette (*Figure 4M*). We then performed electroporation to introduce the plasmids into the hippocampus, after which the embryos were returned to the abdomen. At 7 weeks after birth, mice were fixed using cardiac perfusion of paraformaldehyde (*Figure 4M*). Coronal cryosections of the CA1 and 2 layers in the dorsal hippocampus were cut and the neuronal morphology was analyzed using z stacks of laser confocal microscopy images. Since the cytosolic GFP labels both dendrites and axons, we counted the number of neurites (including both dendrites and axons) that directly emerged from the soma. Consistent with the knockout results, we found that knockdown of Drp1$_{ABCD}$ significantly increased the number of neurites, compared to the scramble control, in the hippocampus in vivo (*Figure 4N*).

## The loss of Drp1$_{ABCD}$ increases sensorimotor gating function

To test whether the loss of Drp1$_{ABCD}$ affects brain function, behavioral phenotypes were systematically characterized in control and Drp1$_{exonA}$-KO mice. We observed normal general locomotor activities in Drp1$_{exonA}$-KO mice in open field test (*Figure 4—figure supplement 3A*). Intriguingly, KO mice exhibited significantly increased prepulse inhibition (PPI) of the acoustic startle without alterations in the startle response (*Figure 4O and P*). PPI, as a measure of sensorimotor gating, involves several brain regions (including the hippocampus, medial prefrontal cortex, amygdala, and nucleus accumbens) (*Lee and Davis, 1997*; *Swerdlow et al., 2001*). Sensorimotor gating function enables selective attention that distinguishes or separates critical information from background noise. In humans, sensorimotor gating function is often referred to as the cocktail party effect, which allows one to talk with someone even in a crowded party environment (*Lee and Davis, 1997*; *Swerdlow et al., 2001*). This gating function is important for human health and its defects have been associated with mental illness, such as schizophrenia and autism spectrum disorders (*Lee and Davis, 1997*; *Swerdlow et al., 2001*). At this moment, we do not know the exact mechanistic basis underlying this enhanced sensorimotor gating in Drp1$_{exonA}$-KO mice; however, the increased number of dendrites or the decreased postsynaptic endocytosis in Drp1$_{exonA}$-KO mice may contribute to the enhancement in sensorimotor gating function. Behavioral changes in Drp1$_{exonA}$-KO mice appeared to be specific to sensorimotor gating since we observed no alterations in spatial working and recognition memory tasks in Y-maze tests (*Figure 4—figure supplement 3B*), motor coordination in rotarod test (*Figure 4—figure supplement 3C*), and anxiety level in elevated plus maze test (*Figure 4—figure supplement 3D*).

In summary, we found, for the first time, that the novel brain-specific isoform Drp1$_{ABCD}$ controls postsynaptic endocytosis independently of mitochondrial division. It would be important to test if this, in turn, results in the accumulation of cargoes on the postsynaptic surface and leads to ectopic formation of dendrites in future studies. Since the expression of Drp1$_{ABCD}$ is induced during the

postnatal period, Drp1$_{ABCD}$ may control the number of dendrites by suppressing unwanted, excess dendrite formation in neuronal network wiring during postnatal brain development.

# Materials and methods

## Key resources table

| Reagent type (species) or resource | Designation | Source or reference | Identifiers | Additional information |
|---|---|---|---|---|
| Genetic reagent (*M. musculus*) | Wild-type mice | This paper | | Materials and methods: Generation of Drp1$_{exonA}$-KO mice using CRISPR/Cas9 |
| Genetic reagent (*M. musculus*) | Drp1$_{exonA}$-KO mice | This paper | | Materials and methods: Generation of Drp1$_{exonA}$-KO mice using CRISPR/Cas9 |
| Genetic reagent (*M. musculus*) | Thy1-GFP-M transgenic mice | Jackson Laboratory | Stock #: 007788 | |
| Genetic reagent (*M. musculus*) | C57BL/6J mice | Jackson Laboratory | Stock #: 000664 | |
| Cell line (*M. musculus*) | WT and Drp1-KO MEFs | *Kageyama et al. (2014)* | | |
| Antibody | Rabbit polyclonal anti-exon AB | *Itoh et al. (2018)* | | WB (1:2000) |
| Antibody | Mouse monoclonal anti-Psd-95 | EMD Millipore | Cat #: MABN68 | WB (1:2000) |
| Antibody | Mouse monoclonal anti-pan-Drp1 | BD Biosciences | Cat #: 611113 | WB (1:2000) |
| Antibody | Mouse monoclonal anti-PDH subunit E2/E3bp | Abcam | Cat #: ab110333 | IF (1:300) |
| Recombinant DNA reagent | HA-Drp1$_{ABCD}$ | *Itoh et al. (2018)* | | |
| Recombinant DNA reagent | HA-Drp1$_{BCD}$ | *Itoh et al. (2018)* | | |
| Recombinant DNA reagent | Psd95.FingR-GFP | Addgene | Cat #: 46295 | *Gross et al. (2013)* |
| Recombinant DNA reagent | mCherry-Clathrin (CLC) | Addgene | Cat #: 27680 | |
| Recombinant DNA reagent | Psd-95-mCherry | *Blanpied et al. (2008)* | | |
| Recombinant DNA reagent | GFP-GluR1 | *Hussain et al. (2014)* | | |
| Recombinant DNA reagent | pSUPER-Scramble | This paper | | Materials and methods: Plasmids |
| Recombinant DNA reagent | pSUPER-AB | This paper | | Materials and methods: Plasmids |
| Recombinant DNA reagent | pSUPER-GFP-Scramble | This paper | | Materials and methods: Plasmids |
| Recombinant DNA reagent | pSUPER-GFP-AB | This paper | | Materials and methods: Plasmids |
| Recombinant DNA reagent | pCAGGS1-Drp1$_{ABCD}$ (resistant form) | This paper | | Materials and methods: Plasmids |
| Chemical compound, drug | Dynasore hydrate | Sigma-Aldrich | Cat #: D7693 | |
| Chemical compound, drug | NMDA | Tocris | Cat #: 0114 | |

*Continued on next page*

*Continued*

| Reagent type (species) or resource | Designation | Source or reference | Identifiers | Additional information |
|---|---|---|---|---|
| Chemical compound, drug | Glycine | Tocris | Cat #: 0219 | |
| Chemical compound, drug | Tetrodotoxin (TTX) | Tocris | Cat #: 1078 | |

## Generation of Drp1$_{exonA}$-KO mice using CRISPR/Cas9

All animal work was conducted according to the guidelines established by the Johns Hopkins University Committee on Animal Care and Use. To engineer the mouse Dnm1l gene that encodes Drp1, sgRNA-encoding sequences (5'- AAAATGGTAAATTTCAGAGC- 3' to target inside the A exon and 5'-TAAAAAGTTGATTGGTGAAT- 3' to target downstream of the A exon) were cloned into the BbsI site of pX330-T7 and amplified from pX330-T7 with a leading T7 promoter by PCR (*Igarashi et al., 2018*). These sgRNAs were in vitro transcribed using the HiScribe T7 Quick High Yield RNA Synthesis Kit (New England Biolabs) and purified using the MEGAclear Kit (Ambion). Cas9 mRNA was in vitro transcribed using NotI-linearized pX330-T7 and the mMESSAGE mMACHINE T7 Ultra Kit (Ambion) and purified by LiCl precipitation. Pronuclear injections of zygotes from B6SJLF1/J mice (Jackson Laboratory, stock no. 100012) were performed at the Johns Hopkins University Transgenic Facility using a mix of Cas9 mRNA and two sgRNA-encoding sequences in injection buffer (10 mM Tris-HCl, 0.1 mM EDTA filtered with 0.2 μm pore size). Three combinations of concentrations were used: 100 ng/μl of Cas9 mRNA and 50 ng/μl of each sgRNA, 100 ng/μl Cas9 of mRNA and 25 ng/μl of each sgRNA, and 25 ng/μl of Cas9 mRNA and 12.5 ng/μl of each sgRNA. The embryos were cultured at 37°C in the $CO_2$ incubator for 2 hr and then transferred into the oviducts of pseudopregnant ICR females (25 embryos per mouse) (Envigo). Sixteen pups were obtained and their genotypes were analyzed by PCR using the following primers: 5'-AGACCTCTCATTCTGCAGCT-3' and 5'-GTGGATGGTCGCTGAGTTTG-3'. We identified one founder mouse that truncated 96 bp to remove the A exon, resulting in A knock out. The A exon (KFQSWN) was replaced with 20 amino acids (KWEIIAIAKSEIFRIGINI) and a stop codon. By breeding with the Thy1-GFP-M transgenic mouse line (Jackson Laboratory, stock no. 007788), we generated Thy1-GFP/homozygous Drp1$_{exonA}$-KO mice and Thy1-GFP/wild-type mice.

## Plasmids

To create the HA-Drp1$_{BCD}$ plasmid, Drp1$_{ABCD}$ in the HA-Drp1$_{ABCD}$ plasmid (*Itoh et al., 2018*) was replaced with the full length of Drp1$_{BCD}$ at the BamHI/NotI sites. To create the GFP-Drp1$_{ABCD}$ plasmid, (SAGG)$_5$ linker sequence and full-length Drp1$_{ABCD}$ were cloned into the BglII/EcoRI sites and the XhoI/SmaI sites of pEGFP-C1 (Clontech), respectively. Drp1$_{ABCD}$ was replaced with Drp1$_{ACD}$ and Drp1$_{BCD}$ to create the GFP-Drp1$_{ACD}$ and GFP-Drp1$_{BCD}$ plasmids. To generate the shRNA plasmids, the following target sequences were cloned into pSUPER (Oligoengine, VEC-PBS-0002) or pSUPER-GFP (*Yamada et al., 2018*). Scramble: CCTAAGGTTAAGTCGCCCTCGttcaagagaCGAGGGCGACTTAACCTTAGG, AB: ATTTCAGAGCTGGAACCCTGCttcaagagaGCAGGGTTCCAGCTCTGAAAT, and pan-Drp1: GCTTCAGATCAGAGAACTTATttcaagagaATAAGTTCTCTGATCTGAAGC. To generate a knockdown-resistant Drp1$_{ABCD}$ plasmid, both target sequences for AB and pan-Drp1 were replaced to the following resistant form. AB: ATTTCAGAGCTGGAACCCTGC to G̲T̲T̲C̲C̲A̲A̲AG T̲T̲GGAAT̲CC̲AGC, and pan-Drp1: GCTTCAGATCAGAGAACTTAT to G̲T̲T̲GCA̲A̲AT̲T̲C̲G̲C̲G̲AGC T̲G̲AT. Underlined cases are the added silent mutations. Full length of Drp1$_{ABCD}$ with silent mutations was cloned into the XhoI/NotI sites of pCAGGS1 vector.

## Immunoblotting

Mouse tissues were harvested, flash-frozen in liquid nitrogen, and homogenized in RIPA buffer (Cell Signaling Technology, 9806) that contained cOmplete Mini Protease Inhibitor (Roche, 11836170001). Lysates were centrifuged at 14,000 x g for 10 min at 4°C and the supernatants were collected. Proteins were separated by SDS–PAGE and transferred onto Immobilon-FL membranes (Millipore). The antibodies used were exon AB (*Itoh et al., 2018*), pan-Drp1 (BD Biosciences, 611113), PDH subunit E2/E3bp (Abcam, ab110333), GAPDH (Thermo, MA5-15738), actin (Santa

Cruz Biotechnology, sc-1615), Psd-95 (EMD Millipore, MABN68), clathrin (BD Biosciences, 610499), beta-III tubulin (Abcam, ab18207), GFP (Molecular probe, A11121), GluR1 (EMD Millipore, MAB397), GluR2 (*Araki et al., 2010*) and GluR3 (*Araki et al., 2010*). Immunocomplexes were visualized using fluorescently-labeled secondary antibodies and detected using a PharosFX Plus Molecular Imager (Bio-Rad).

## Neuronal cultures and immunofluorescence microscopy

Hippocampal neurons were isolated and cultured in vitro as previously described (*Araki et al., 2015*). In brief, E18.5 embryos were decapitated, and brains were quickly removed and transferred in cold Dissection media [1 x HBSS (Gibco, 14185052), 1 mM sodium pyruvate (Gibco, 11360070), 10 mM HEPES (Gibco, 15630080), 30 mM glucose, 100 U/ml penicillin, and 100 µg/ml streptomycin]. Hippocampi were dissected under a binocular microscope and incubated in Dissection medium supplemented with 0.5 mg/ml papain (Worthington, LS003119) and 0.01% DNase (Sigma, DN25) for 20 min at 37°C. Hippocampi were washed once with warm Neurobasal medium (Gibco, 21103049) supplemented with 100 U/ml penicillin, 100 µg/ml streptomycin, 2 mM GlutaMax (Gibco, 35050061), 2% B-27 (Gibco, 17504044) and 5% horse serum (Gibco, 26050088). Neurons were triturated and plated on 18 mm poly-L-lysine-coated coverslips at a density of 160,000 cells/well in 12-well tissue culture plates in 1 ml of the Neurobasal medium supplemented with 100 U/ml penicillin, 100 µg/ml streptomycin, 2 mM GlutaMax, 2% B-27% and 5% horse serum. After 24 hr, neurons were switched and maintained thereafter in Neurobasal media with 2 mM GlutaMax and 2% B-27. Cultured neurons were fed with half-media changes once per week. Cells were transfected with Lipofectamin 2000 (Invitrogen) in accordance with the manufacturer's manual. After 2–3 days, neurons were fixed using PBS containing 4% paraformaldehyde, washed in PBS, permeabilized with 0.2% Triton X-100/PBS, and blocked in 0.5% BSA/PBS (*Adachi et al., 2016*). The cells were incubated with antibodies to pan-Drp1, HA (Novus Biologicals, NB600-362), RFP (antibodies-online, ABIN129578), VGLUT1 (Synaptic systems, 135304), MAP2 (Thermo Fisher, MA5-12826) and PDH subunit E2/E3bp, followed by the appropriate secondary antibodies. Samples were mounted in Prolong Gold Antifade Reagent (Cell Signaling, 9071) and viewed using Zeiss LSM510-Meta, LSM700 FCS, and LSM800 GaAsP laser scanning confocal microscopes. To determine the size of the mitochondria in the dendrites, we first examined serial confocal images along the Z-axis to identify individual mitochondria and then measured their length using ImageJ.

## PSD fractionation

Fractionation of post-synaptic density was performed as described previously (*Araki et al., 2015*). In brief, mouse whole brain was dissected and homogenized by a dounce homogenizer 30 times in Buffer A (0.32 M sucrose, 10 mM Hepes, pH7.4, with cOmplete Mini Protease Inhibitor). The homogenate was centrifuged at 1000 x g for 10 min at 4°C. The post-nuclear supernatant was collected and centrifuged at 13,800 x g for 20 min at 4°C. The supernatant was kept as S2 fraction. The pellet was resuspended in 3 volumes of Buffer A (P2 fraction). The P2 fraction was layered onto a discontinuous sucrose gradient (0.85, 1.0, and 1.4 M) in 10 mM Hepes (pH7.4) with cOmplete Mini Protease Inhibitor and centrifuged at 82,500 x g for 2 hr at 4°C. The interface between 1.0 and 1.4 M sucrose was collected as the synaptosome fraction (Syn) and diluted with 80 mM Tris-HCl (pH 8.0). An equal volume of 1% Triton X-100 was added and rotated for 10 min at 4°C, then centrifuged at 32,000 x g for 20 min. The supernatant was collected as Triton-soluble synaptosome (Syn/Tx) fraction, and the pellet was resuspended in 80 mM Tris-HCl (pH 8.0) (PSD fraction).

## Electron microscopy

Cultured neurons were fixed with 2% glutaraldehyde, 3 mM $CaCl_2$, and 0.1 M cacodylate buffer, pH 7.4, for 1 hr. After washes, samples were post-fixed in 2.7% $OsO_4$ and 167 mM cacodylate, pH 7.4, for 1 hr on ice (*Kageyama et al., 2014*; *Wakabayashi et al., 2009*). After washes in water, samples were incubated in 2% uranyl acetate for 30 min. After dehydration using 50, 70, 90, and 100% ethanol, samples were embedded in EPON resin. Ultrathin sections were obtained using a Reichert-Jung ultracut E, stained with 2% uranyl acetate and 0.3% lead citrate, and viewed using a transmission electron microscope (H-7600; Hitachi) equipped with a dual CCD camera (Advanced Microscopy Techniques).

For dynasore treatment, cells were incubated with 80 µM of dynasore (Sigma-Aldrich, D7693) in culture medium for different times, then fixed and further processed for electron microscopy as described above. To stimulate endocytosis through chemical long-term depression (chemical LTD), neurons were incubated with 20 µM of NMDA (Tocris, 0114), 10 µM of glycine (Tocris, 0219), 0.3 mM of $MgCl_2$, 2 mM of $CaCl_2$ and 1 µM of TTX (Tocris, 1078) in Base buffer (10 mM HEPES, pH 7.4, 140 mM NaCl, 2.4 mM KCl, 10 mM glucose) for 4 min. As a control, Base buffer containing 2 mM of $MgCl_2$, 2 mM of $CaCl_2$ and 1 µM of TTX was used. To induce chemical LTD in the presence of dynasore, neurons were first incubated for 1 min with 80 µM of dynasore in the culture medium and followed by a 3 min chemical LTD treatment in the presence of dynasore (80 µM). Neurons were then fixed and processed as described above.

### Analysis of endocytic zone

Hippocampal neurons (DIV22) were transfected with 1 µg of Psd-95.FingR-GFP plasmids (Addgene, 46295) and 250 ng of mCherry-clathrin light chain plasmids (Addgene, 27680) per coverslip in 12-well plates. Two days after transfection, neurons were treated with chemical LTD stimulation, fixed in PBS containing 4% formaldehyde and 4% sucrose for 20 min, washed with PBS and mounted. Neurons were selected based on GFP fluorescence, and mCherry/GFP images were taken. Images were acquired with LSM800 GaAsP laser scanning confocal microscopes and analyzed using ImageJ.

### Surface biotinylation assay

Cultured neurons were washed once with Base buffer containing 2 mM $MgCl_2$ and 2 mM $CaCl_2$ at room temperature; they were then washed twice with an ice-cold version of the same buffer. Cell-surface proteins were biotinylated with 1 mg/mL sulfo-NHS-SS-biotin (Pierce, 21331) in the same buffer for 20 min on ice. The remaining biotin was quenched by washing the cells two times for 5 min each with ice-cold PBS containing 20 mM glycine, 2 mM $MgCl_2$ and 2 mM $CaCl_2$. Immediately after quenching, the neurons were washed twice with PBS containing 2 mM $MgCl_2$ and 2 mM $CaCl_2$ and then lysed with RIPA buffer that contained cOmplete Mini Protease Inhibitor. The biotinylated cell surface proteins were precipitated using NeutrAvidin agarose (Pierce, 29200). The precipitated proteins and total cell lysates were separated by SDS-PAGE and blotted with antibodies to GluR1, GluR2, GluR3 and GAPDH.

### GluR1 internalization assay

Cultured neurons were transfected with 1 µg of Psd-95-mCherry plasmids (*Blanpied et al., 2008*) and 1 µg of GFP-GluR1 plasmids (*Hussain et al., 2014*) per coverslip in 12-well plates. Two days after transfection, the neurons were treated with chemical LTD stimulation, fixed in PBS containing 4% formaldehyde and 4% sucrose for 8 min, washed with PBS and blocked in 1% BSA/PBS for 30 min. To label surface GFP-GluR1, the cells were incubated with GFP antibody (*Senoo et al., 2019*) at 4°C overnight and then treated with Alexa Fluor 647-conjugated secondary antibodies. Images were acquired using LSM800 GaAsP laser scanning confocal microscopes and analyzed using ImageJ. Identical settings were used to acquire each image within an experiment.

### Transferrin uptake

MEFs were incubated with 5 µg/ml of Alexa-Fluor-647-transferrin (Thermo, T23366) in the culture medium for 30 min at 4°C or 37°C. Cells were washed twice with cold PBS, fixed using PBS containing 4% paraformaldehyde, washed in PBS and visualized by confocal microscopy. Mean fluorescent signals in each cell were measured using Image J. Cultured neurons were incubated with 50 µg/ml of FITC-transferrin (Thermo, T2871) in the culture medium for 15 min at 4°C or 37°C. Cells were washed twice with cold PBS, fixed using PBS containing 4% paraformaldehyde and 4% sucrose, washed in PBS and then visualized by confocal microscopy. Mean fluorescent intensity was measured along dendrites (100 µm in length) using Image J.

### In utero electroporation

In utero electroporation that targeted the dorsal hippocampus region was performed according to our published protocol with some modifications (*Saito et al., 2016*). Pregnant mice (C57BL/6J, The Jackson Laboratory, stock no. 000664) were anesthetized at embryonic day 15.5 (E15.5) by

intraperitoneal administration of a mixed solution of ketamine HCl (100 mg/kg), xylazine HCl (7.5 mg/kg), and buprenorphine HCl (0.05 mg/kg). After the uterine horn was exposed by laparotomy, the CAG promoter-driven eGFP expression plasmid, pCAGGS1-eGFP (1 µg/µl), together with the Drp1$_{ABCD}$ knockdown plasmid, pSUPER-AB (1 µg/µl), was injected (1–2 µl) into the lateral ventricles with a glass micropipette made from a microcapillary tube (Narishige, Cat #GD-1). Using a ø3mm electrode (Nepagene #CUY650P3), the plasmids were delivered into the dorsal hippocampus by electric pulses (40V; 50 ms), which were charged four times at intervals of 950 ms with an electroporator (Nepagene #CUY21EDIT). After electroporation, the uterine horn was replaced in the abdominal cavity to allow the embryos to continue to develop.

## Behavioral analysis

All of the behavior tests were performed in mice of 2–5 months of age at the Behavior Core of the Johns Hopkins University School of Medicine. For open field tests, mice were placed in a Photobeam Activity System Open Field (San Diego Instruments, CA, USA) and their movement was recorded for 30 min (*Breu et al., 2016*). The open field chamber consisted of a clear Plexiglas box (40 × 40 × 37 cm) with 16 horizontal and 16 vertical photo-beams to assess locomotion and location tendency. Activity parameters were quantified as the number of beam breaks.

For PPI tests, mice were put in a clear Plexiglas cylinder (3.8 cm in diameter) within a startle chamber (San Diego Instruments) and tested for their sensorimotor gating function using SR-LAB software (*Nasu et al., 2014*; *Saito et al., 2016*) (Startle Response System, San Diego Instruments, CA, USA). A loudspeaker mounted 24 cm above the cylinder provided acoustic stimuli and background noise (70 dB) and controlled the delivery of all stimuli to the animal by SR-LAB software and the interface system. A maximum voltage during the 100 ms period beginning at the stimulus onset was measured as a startle amplitude. To initiate the test, mice were given a 5 min acclimation period with 70 dB background noise; this background noise was present throughout the entire session. After acclimation, mice were exposed to a pulse (a 120 dB, 40 ms) 10 times and then the background-only session 10 times at a 20 s inter-stimuli interval (habituation session). In experimental sessions, mice were exposed to the following types of trials: pulse alone trial (a 120 dB, 100 ms broadband burst); the omission of stimuli (no pulse, only background noise); and five prepulse-pulse combination trials. Broadband bursts (20 ms) were individually presented as prepulses for 80 ms before the pulse (120 dB, 100 ms broadband pulse). Each session consisted of six presentations of each type of trial presented at a 20 s inter-stimulus interval in a pseudorandom order. PPI was defined as a reduced percentage of startle amplitude in prepulse-pulse trials compared to the startle amplitude in startle-alone trials.

For the Y-maze test, mice were placed in a Y-shaped maze with three arms (38 × 7.5 × 12 cm) at 120-degree angles from each other. After introduction to the center of the maze, mice are allowed to freely explore the three arms and are video-recorded for 10 min. The number of arm entries and the time spent in each arm were scored in order to calculate the percentage of alternation.

For rotarod tests, mice were placed on the rod spindle assembly (3.0 cm in diameter) of the Rotamex-5 system (*Kageyama et al., 2012*) (Columbus Instruments, OH, USA). Mice were first trained at 4.0 rpm for 5 min. After this training session, the rotarod was accelerated with a 1.0 rpm increase in rotational speed every 5 s. The time elapsed before falling was recorded for each mouse. Three consecutive trials were performed and the results were averaged in each mouse.

For the elevated plus maze test, a mouse was placed on the starting platform in the plus maze (San Diego Instruments Inc, San Diego, CA, USA) and the mouse's behaviors were video-recorded for 5 min. We scored the numbers of entries into the closed and open arms and the time spent in the closed and open arms.

## MEFs and lentiviruses

Drp1-KO MEFs were cultured in Iscove's modified Dulbecco's medium supplemented with 10% fetal bovine serum as described previously (*Kageyama et al., 2014*). Genotypes of MEFs were confirmed by PCR as described (*Kageyama et al., 2014*). No contamination of mycoplasma has been confirmed. Lentiviruses were produced as described previously (*Itoh et al., 2018*).

## Acknowledgements

We thank past and present members of the Iijima and Sesaki labs for helpful discussions and technical assistance. We also thank Mr. Michael Delannoy and Dr. Yuuta Imoto for the technical support of EM work and valuable discussions. We are grateful to Dr. Thomas Blanpied for the Psd-95-mCherry plasmid. This work was supported by NIH grants to MI (GM131768), HS (GM123266 and GM130695), AK and MVP (DA041208) and MVP (MH083728).

## Additional information

### Funding

| Funder | Grant reference number | Author |
|---|---|---|
| National Institute of General Medical Sciences | GM131768 | Miho Iijima |
| National Institute of General Medical Sciences | GM123266 | Hiromi Sesaki |
| National Institute of General Medical Sciences | GM130695 | Hiromi Sesaki |
| National Institute on Drug Abuse | DA041208 | Mikhail Pletnikov Atsushi Kamiya |
| National Institute of Mental Health | MH083728 | Mikhail Pletnikov |

The funders had no role in study design, data collection and interpretation, or the decision to submit the work for publication.

### Author contributions

Kie Itoh, Conceptualization, Investigation, Writing—original draft, Writing—review and editing; Daisuke Murata, Yoichi Araki, Atsushi Saito, Yoshihiro Adachi, Shuo Li, Data curation; Takashi Kato, Investigation; Tatsuya Yamada, Atsushi Igarashi, Formal analysis; Mikhail Pletnikov, Richard L Huganir, Shigeki Watanabe, Atsushi Kamiya, Supervision; Miho Iijima, Hiromi Sesaki, Conceptualization, Resources, Supervision, Funding acquisition, Writing—original draft, Project administration, Writing—review and editing

### Author ORCIDs

Kie Itoh (iD) https://orcid.org/0000-0003-4379-400X
Hiromi Sesaki (iD) https://orcid.org/0000-0002-6877-3929

### Ethics

Animal experimentation: This study was performed in strict accordance with the recommendations in the Guide for the Care and Use of Laboratory Animals of the National Institutes of Health. All animal work was conducted according to the guidelines established by the Johns Hopkins University Committee on Animal Care and Use. The protocol (MO17M181) has been approved by the same committee.

### Decision letter and Author response

Decision letter https://doi.org/10.7554/eLife.44739.020
Author response https://doi.org/10.7554/eLife.44739.021

## Additional files

### Supplementary files

• Transparent reporting form DOI: https://doi.org/10.7554/eLife.44739.019

**Data availability**

All data generated or analysed during this study are included in the manuscript and supporting files.

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
