## [Decision Letter]

[Editors’ note: this article was originally rejected after discussions between the reviewers, but the authors were invited to resubmit after an appeal against the decision.]

Thank you for submitting your work entitled "Brain-specific Drp1 regulates postsynaptic endocytosis and dendrite formation independently of mitochondrial division" for consideration by *eLife*. Your article has been reviewed by three peer reviewers, and the evaluation has been overseen by a Reviewing Editor and a Senior Editor. The following individuals involved in review of your submission have agreed to reveal their identity: David Perrais (Reviewer #1); Mary Kay Lobo (Reviewer #3).

Our decision has been reached after consultation between the reviewers. Based on these discussions and the individual reviews below, we regret to inform you that your work will not be considered further for publication in *eLife*.

We should stress that all of the reviewers found your study very interesting. However, they also all agreed that it was still unclear whether or not brain-specific Drp1 is directly involved in postsynaptic endocytosis. As you know, it is *eLife*'s policy to reject papers where additional work is required to validate major conclusions, when that work would take more than two months. The reviewers have suggestions for further experiments, which we hope you will find useful, but they acknowledged that these experiments would take longer than two months.

Reviewer #1:

The manuscript by Itoh et al. describe the consequences of the deletion of a brain specific splice variant of Drp1 termed Drp1AB. Drp1 is a well characterized GTPase which belongs to the dynamin-like protein family. It has been involved mainly in the fission reaction of mitochondria. In addition, the groups of Miho Iijima and Hiromi Sesaki, corresponding authors of the present manuscript, have very recently found (Itoh et al., 2018) that the splice variant Drp1_ABCD_, with all four additional inserts, is specific for the brain and is present not only at mitochondria scission sites but also at late endosomes and lysosomes, and the plasma membrane. In the present study, they examine the effect of deleting specifically Drp1_ABCD_ (or Drp1_AB_ deletion, because the only variant containing both A and B also contains C and D) with a Drp1_exonA_ KO mouse and shRNA targeting specifically Drp1 mRNAs containing A and B exons. They found, using cultured hippocampal neurons, that Drp1_ABCD_ is present in dendritic spines and that its absence increases the number of CCPs near post-synaptic sites, suggesting that Drp1_ABCS_ affects clathrin mediated endocytosis. Surprisingly, spine density was not affected but the number of primary dendrites was increased in cultured neurons and in pyramidal neurons in cortex and hippocampus in vivo. Finally, KO mice have a deficit in two behavioral tests involving sensory gating, but not in classical tests of locomotor activity, anxiety, working memory.

The experiments are in general well performed and the results are clearly and concisely presented. The results are interesting but it is hard to see clear connections between the three main effects of Drp1_AB_ deletion on post-synaptic CCPs, primary dendrites and the modulation of the startle response. In particular, before further consideration for publication, it is important to determine if and how post-synaptic endocytosis is affected and if there is a direct connection with the number of primary dendrites.

Therefore I propose to:

1) Test directly the possibility that postsynaptic endocytosis is affected. First, the number of CCPs near PSDs can be quantified (as in Blanpied et al., 2002, Lu et al., 2007 or Rosendale et al., 2017). Second, test the distribution and internalization of post-synaptic cargoes (e.g. transferrin and AMPARs are the classical ones).

2) Reevaluate the effect of dynasore. It is described as affecting dynamin at a late stage of endocytosis. However, dynasore also affects Drp1, at least in vitro (Figure 2G of Macia et al., 2006) so a different dynamin (or clathrin) blocker such as mitmab or pitstop2 should be used to test if the effects goes through dynamin or Drp1 inhibition.

3) Explain further in what sense the approach with shRNA is more specific than the Drp1_exonA_-KO. Do the authors mean that they delete the exonA containing Drp1 mRNA in target neurons, and not in the other cells types (e.g. astrocytes)?

4) Explain why the neurons in culture were transfected at an age that seems quite old (22-24 DIV). At this age, cell survival and transfections are notoriously poor. The authors should indicate if and how often medium was replaced in culture. Why did the authors choose such an old age? Is the effect of Drp1_AB_ loss only seen after this time in culture? The authors should document at least in Drp1_exonA_ KO neurons if the number of primary dendrites is affected at early stages of neuronal development. Is there any other example where the number of primary dendrites can be affected late in neuronal development?

5) Figure 1C: the post-natal expression of Drp1_AB_, is very low at P8 (unlike PSD95 as written in the text). This could be in line with my previous point, if the effects of Drp1_AB_ deletion are only visible at late stages. Could the authors comment this further?

Reviewer #2:

This is an interesting study that describes the involvement of one particular splice variant (isoform) of Drp1 in postsynaptic clathrin-mediated endocytosis (CME), which ultimately affects neuronal morphology and animal behavior. While the experiments are well-executed and the results mostly convincing, the study primarily addresses "phenomena" from 36K feet up without attempting to delve into the exact role of this Drp1 variant here, or its mechanism. Therefore, this reviewer finds the study to be largely descriptive and observational, without being able to infer roles or mechanisms. Several important questions therefore need to be addressed.

Specifically, what is Drp1 doing here while in the presence of dynamin (which mediates fission one would think), and associated BAR domain-containing proteins, which presumably function to create the membrane invaginations in the first place? How does this Drp1 variant contribute to membrane remodeling specifically in the context of postsynaptic CME, or does it? Do other modes or locations of endocytosis rely on this variant? Is this Drp1 variant recruited to the postsynaptic endocytic pit, or to the PM in general, in the absence of dynamin? Is there a role for Mff in the selective PM recruitment of this variant as alluded by Li et al., 2013? It is not clear what allows the selective recruitment of this Drp1 splice variant to the PM.

The recruitment and role of this variant should therefore be further investigated in the context of isoform-specific and pan-dynamin knockdown, which should provide insight into Drp1's role and mechanism, specifically in postsynaptic CME. What is the morphology of these endocytic pits in the absence (or presence) of dynamin 1/2/3? Can Drp1 substitute for dynamin in postsynaptic CME? Further, are there specific binding partners at the PM for this variant versus other Drp1 isoforms? In other words, what is the basis of the selective recruitment of this Drp1 splice variant over others?

In a nutshell, the following sentence from this manuscript requires additional clarification: "The accumulation of shallow and U-shaped CCPs, but not omega-shaped

ones, suggest that Drp1_ABCD_ may function at an early step upstream of the constriction and severing of the neck of coated pits that is mediated by dynamin (Schmid and Frolov, 2011)."

What is the role and mechanism of Drp1 here? This should be established to complete this manuscript, which will provide critical insights into the largely unstudied role of Drp1, outside of the context of mitochondria.

Reviewer #3:

The study by Itoh et al. investigates the function of the Drp1-_ABCD_ isoform, which contains all four alternative exons and is specific to the brain. The authors use both shRNA knockdown and CRISPR isoform specific knockouts. They show that Drp1-_ABCD_ is expressed during postnatal development, is enriched in dendritic spines, and regulates postsynaptic clathrin-mediated endocytosis. Drp1-_ABCD_ KO or KD induces primary neuron dendrites in culture and in utero KD in hippocampus increases primary neurites. Finally, Drp1-_ABCD_ KO mice display enhanced sensorimotor gating behavior in mice. This is a clear and well written study that provides novel information into Drp1-_ABCD_ function. However, additional studies would further support the findings and rule out a role for Drp1-_ABCD_ in mitochondria function.

1) Please include representative images used to quantify Spine/Dendrite expression of HA-Drp1 -BCD (Figure 1E and F) similar to those shown of HA-Drp1-_ABCD_ in Figure 1G.

2) The mitochondria analysis in Figure 3 and S2 is preliminary. Since the main role of Drp1 is to mediate mitochondrial fission it will be important to determine if mitochondria are unaltered with Drp1-_ABCD_-KO and KD. The authors should provide quantification of mitochondria morphology, which may require genetic labeling of mitochondria. Additionally, Drp1 can alter mitochondria across dendritic regions. From the images it seems that only proximal dendrites are examined. It will be important to analyze the mitochondria across the dendritic field.

3) In the last sentence of the Abstract the authors state that their results suggest that DRP1-_ABCD_ controls morphology and function through surface trafficking of neurotransmitter receptors at postsynapses. Inclusion of more details throughout the manuscript that relate the endocytosis findings to receptor trafficking would help support this conclusion and make it understandable to readers unfamiliar with the neurotransmitter receptor trafficking field.

[Editors’ note: what now follows is the decision letter after the authors submitted for further consideration.]

Thank you for choosing to send your work entitled "Brain-specific Drp1 regulates postsynaptic endocytosis and dendrite formation independently of mitochondrial division" for consideration at *eLife*. Your letter of appeal has been considered by a Senior Editor and a Reviewing editor, and we are prepared to consider a revised submission.

We were helped in making our decision by the original reviewers, who read your revised manuscript. They all agreed that your changes had strengthened the story, and thought it would be helpful to include an illustration or model demonstrating how Drp1 _ABCD_ might function in postsynaptic endocytosis.

The reviewers also suggested a couple of additional experiments. First, there seems to be a discrepancy between the EM data in Figure 2Q, showing more CCPs at the postsynaptic terminal under basal conditions in the KO cells, and the light microscopy data in Figure 3 A-B, which appears to show less clathrin at the postsynaptic terminal terminal. The reviewer who pointed this out felt that you needed more direct evidence for your hypothesis that Drp1_ABCD_ regulates cargo recycling and suggested that you investigate the localisation of AMPA receptors in your KO cells.

The other suggested experiment was to carry out a similar study to the one in supplementary Figure 6, showing a lack of effect of dynasore on mitochondrial morphology in MEFs, but this time using neurons as in Figure 3D. Because the labelled mitochondria often overlap, it will be important to show how you measure your mitochondrial length. Both of these experiments appear to be straightforward enough to complete within two months.

[Editors' note: further revisions were requested prior to acceptance, as described below.]

Thank you for resubmitting your work entitled "Brain-specific Drp1 regulates postsynaptic endocytosis and dendrite formation independently of mitochondrial division" for further consideration at *eLife*. Your revised article has been favorably evaluated by Didier Stainier (Senior Editor), a Reviewing Editor, and three reviewers.

The manuscript has been improved but there are some remaining issues that need to be addressed before acceptance.

Specifically, there is still the issue of the connection between the phenotypes, which has not been solved. The authors have convincing data regarding the effect of Drp1_ABCD_ deletion on post-synaptic endocytosis and the formation of ectopic dendrites. However, they do not show that ectopic dendrite formation occurs because postsynaptic endocytosis is affected. Therefore, the last sentence of the Abstract and the last paragraph of the Discussion must be changed.

---

## [Author Response]

[Editors’ note: this article was originally rejected after discussions between the reviewers, but the authors were invited to resubmit after an appeal against the decision.]

We should stress that all of the reviewers found your study very interesting. However, they also all agreed that it was still unclear whether or not brain-specific Drp1 is directly involved in postsynaptic endocytosis. As you know, it is eLife's policy to reject papers where additional work is required to validate major conclusions, when that work would take more than two months. The reviewers have suggestions for further experiments, which we hope you will find useful, but they acknowledged that these experiments would take longer than two months.Reviewer #1:The manuscript by Itoh et al. describe the consequences of the deletion of a brain specific splice variant of Drp1 termed Drp1AB. Drp1 is a well characterized GTPase which belongs to the dynamin-like protein family. It has been involved mainly in the fission reaction of mitochondria. In addition, the groups of Miho Iijima and Hiromi Sesaki, corresponding authors of the present manuscript, have very recently found (Itoh et al., 2018) that the splice variant Drp1_ABCD_, with all four additional inserts, is specific for the brain and is present not only at mitochondria scission sites but also at late endosomes and lysosomes, and the plasma membrane. In the present study, they examine the effect of deleting specifically Drp1_ABCD_ (or Drp1AB deletion, because the only variant containing both A and B also contains C and D) with a Drp1_exonA_ KO mouse and shRNA targeting specifically Drp1 mRNAs containing A and B exons. They found, using cultured hippocampal neurons, that Drp1_ABCD_ is present in dendritic spines and that its absence increases the number of CCPs near post-synaptic sites, suggesting that Drp1_ABCS_ affects clathrin mediated endocytosis. Surprisingly, spine density was not affected but the number of primary dendrites was increased in cultured neurons and in pyramidal neurons in cortex and hippocampus in vivo. Finally, KO mice have a deficit in two behavioral tests involving sensory gating, but not in classical tests of locomotor activity, anxiety, working memory.The experiments are in general well performed and the results are clearly and concisely presented. The results are interesting but it is hard to see clear connections between the three main effects of Drp1_AB_ deletion on post-synaptic CCPs, primary dendrites and the modulation of the startle response. In particular, before further consideration for publication, it is important to determine if and how post-synaptic endocytosis is affected and if there is a direct connection with the number of primary dendrites.

We thank this reviewer for the supportive and encouraging comments regarding our manuscript. As described below, we have addressed all of the concerns raised by the reviewer. In particular, we have more clearly shown that postsynaptic endocytosis is decreased in the Drp1_ABCD_-KO neurons due to mislocalization of the endocytic zone from the postsynaptic density.

Therefore I propose to:1) Test directly the possibility that postsynaptic endocytosis is affected. First, the number of CCPs near PSDs can be quantified (as in Blanpied et al., 2002, Lu et al., 2007 or Rosendale et al., 2017). Second, test the distribution and internalization of post-synaptic cargoes (e.g. transferrin and AMPARs are the classical ones).

We are grateful to this reviewer for the constructive suggestions on experiments to further examine the function of Drp1_ABCD_ in endocytosis. As recommended, during the revision process, we tested the postsynaptic positioning of the endocytic zone using mCherry-clathrin light chain and Psd-95- Fibronectin intrabodies in WT and Drp1_exonA_-KO hippocampal neurons. As previously reported (Lu et al., 2007), the majority of mCherry-clathrin signals are localized next to Psd-95 signals in WT neurons (Figure 3A and 3B). In contrast, in the KO neurons, mCherry-clathrin signals were significantly dissociated from Psd-95 signals (Figure 3A and 3B). Therefore, Drp1_ABCD_ is important for localizing endocytic zones to the postsynaptic density. Decreased clathrin amounts near the postsynaptic density may contribute to a decrease in endocytosis in KO neurons (Figure 3C).

To further test the role of Drp1_ABCD_ in endocytosis, we extended our electron microscopic analysis of CCPs in the revised manuscript. In the original manuscript, we showed that the number of CCPs is increased in KO neurons and treatments with a dynamin inhibitor (i.e., dynasore) did not increase the number of CCPs (Figure 2M and N). Conversely, the number of CCPs was lower in WT neurons compared with KO neurons and was increased when treated with dynasore since endocytosis was blocked at the membrane scission step by dynamin inhibition (Figure 2M and N).

Note: As suggested by this reviewer, we reevaluated the effect of dynasore in cells and ruled out that dynasore blocks Drp1 in both neurons and fibroblasts (Figure 2O, 2P and S1). Please see our responses specific to this point below (Comment 3).

In new experiments, we stimulated WT and KO neurons with NMDA. When we stimulated neurons in the presence of dynasore, WT neurons increased the number of postsynaptic CCPs (Figure 2Q). This is likely due to stimulation of endocytosis by NMDA and inhibition of its completion by dynasore. In contrast, KO neurons did not increase the number of CCPs (Figure 2Q). These data suggest that clathrin-mediated endocytosis is still slow in Drp1_ABCD_-KO neurons even when stimulated by NMDA.

Furthermore, when stimulated by NMDA in the absence of dynasore, the number of CCPs was decreased in KO neurons (Figure 2Q). It appears that NMDA induces internalization of some endocytic vesicles in KO neurons. We suggest that KO neurons have slow kinetics of endocytosis but do not completely block it (Figure 2S). From these data, we speculate that Drp1_ABCD_ facilitates the kinetics of endocytosis in early steps, likely through connecting the endocytic zone to the postsynaptic density (Figure 3C). Drp1_ABCD_ does not likely act at the last step of endocytosis such as the scission of the neck of CCPs. This model is also consistent with the accumulation of shallow CCPs, but not omega-shaped CCPs, in the KO neurons (Figure 2L). Finally, Drp1_ABCD_ loss only affected the postsynaptic region and not the presynaptic region, suggesting that Drp1_ABCD_ mainly functions in postsynaptic endocytosis (Figure 2Q and 2R).

As suggested, we also examined endocytosis of FITC-transferrin and found similar uptakes in WT and KO neurons (Figure 3—figure supplement 1A). This is consistent with the previous study (Lu et al., 2007) showing that the dissociation of clathrin and Psd-95 signals does not affect endocytosis of transferrin. We also tried to examine endocytosis of AMPARs using surface biotinylation; however, this was technically very difficult, and we could not detect internalization of GluR1, GluR2 and GluR3 even in WT neurons in response to NMDA stimulation. We will continue to improve our techniques and address this aspect in our future work.

2) Reevaluate the effect of dynasore. It is described as affecting dynamin at a late stage of endocytosis. However, dynasore also affects Drp1, at least in vitro (Figure 2G of Macia et al., 2006) so a different dynamin (or clathrin) blocker such as mitmab or pitstop2 should be used to test if the effects goes through dynamin or Drp1 inhibition.

We agree with this reviewer that it is important to test a potential effect of dynasore on Drp1 in cells. In the revised manuscript, we treated cultured embryonic fibroblasts and hippocampal neurons with dynasore and analyzed mitochondria. We found that dynasore does not change mitochondrial morphology in both cell types (Figure 2O, 2P and S1). As a control for the loss of Drp1 function, Drp1-KO fibroblasts (in which total Drp1 is lost) showed highly elongated mitochondria due to defects in mitochondrial division (Figure 2—figure supplement 1). Therefore, dynasore does not block Drp1 in cells.

3) Explain further in what sense the approach with shRNA is more specific than the Drp1_exonA_-KO. Do the authors mean that they delete the exonA containing Drp1 mRNA in target neurons, and not in the other cells types (e.g. astrocytes)?

Thank you for mentioning this important point. The reasons we used knockdown in addition to knockout is because, as you described, we wanted to test whether the effect of Drp1_ABCD_ loss is cell autonomous in neurons. This is only achieved by the knockdown in a small population of neurons. In addition, the knockdown approach showed that ectopic dendrites can extend 1) for a short period of time (3 days) after knockdown and 2) in matured neurons (3 weeks DIV) with developed dendrites. We added these discussions to the revised manuscript.

4) Explain why the neurons in culture were transfected at an age that seems quite old (22-24 DIV). At this age, cell survival and transfections are notoriously poor. The authors should indicate if and how often medium was replaced in culture. Why did the authors choose such an old age? Is the effect of Drp1AB loss only seen after this time in culture? The authors should document at least in Drp1_exonA_ KO neurons if the number of primary dendrites is affected at early stages of neuronal development. Is there any other example where the number of primary dendrites can be affected late in neuronal development?

Consistent with in vivodata, Western blotting of cultured neurons showed that the expression level of Drp1_ABCD_ increases during the first 0-2 weeks and reaches a plateau at 3-4 weeks (Figure 1D). Based on this expression pattern, we tested the impact of the knockdown and knockout of Drp1_ABCD_ in cultured neurons at 3 weeks.

With regard to our neuron culture condition, we consistently observe similar viability and plasmid transfection efficiency in neurons at 2 and 3 weeks in my laboratory. We can culture neurons with unaffected viability and transfection rate for at least 4 weeks. As requested by this reviewer, we have added more detailed information about our culture conditions to the revised manuscript.

To examine the role of Drp1_ABCD_ in early culture, we knocked down Drp1_ABCD_ at 2 weeks (Figure 4H, 2 weeks). Similar to the 3-week culture, Drp1_ABCD_ knockdown significantly increased the number of primary dendrites in the 2-week culture. These data show that the knockdown effect is reproducible in young neurons at 2 weeks. Interestingly, in this experiment, we found that fewer dendrites were found at 2 weeks compared to 3 weeks—for example, approximately 8 dendrites at 2 weeks and approximately 10 dendrites at 3 weeks in WT neurons (please compare 2 weeks and 3 weeks in Figure 4H). Therefore, neurons likely continue to extend primary dendrites even at 2 weeks in culture. Drp1_ABCD_ may control the continuous morphogenesis of dendrites.

5) Figure 1C: the post-natal expression of Drp1AB, is very low at P8 (unlike PSD95 as written in the text). This could be in line with my previous point, if the effects of Drp1AB deletion are only visible at late stages. Could the authors comment this further?

As this reviewer correctly pointed out, the expression of Drp1_ABCD_ is induced later compared with that of Psd-95 in vivo(Figure 1C) and in vitro(Figure 1D). As described above (Comment 4), new experiments showed that Drp1_ABCD_ depletion induced ectopic extension of dendrites in cultured neurons at 2 weeks (Figure 4H, 2 weeks). Therefore, Drp1_ABCD_ could also function at early stages of neuronal

development.

Reviewer #2:This is an interesting study that describes the involvement of one particular splice variant (isoform) of Drp1 in postsynaptic clathrin-mediated endocytosis (CME), which ultimately affects neuronal morphology and animal behavior. While the experiments are well-executed and the results mostly convincing, the study primarily addresses "phenomena" from 36K feet up without attempting to delve into the exact role of this Drp1 variant here, or its mechanism. Therefore, this reviewer finds the study to be largely descriptive and observational, without being able to infer roles or mechanisms. Several important questions therefore need to be addressed.

We thank this reviewer for the supportive comments that our study is interesting, the experiments are well-executed, and the data are convincing. We have addressed all of the concerns raised by the reviewer as described below.

Specifically, what is Drp1 doing here while in the presence of dynamin (which mediates fission one would think), and associated BAR domain-containing proteins, which presumably function to create the membrane invaginations in the first place? How does this Drp1 variant contribute to membrane remodeling specifically in the context of postsynaptic CME, or does it? Do other modes or locations of endocytosis rely on this variant?

We agree with this reviewer that it is important to more deeply understand the function of Drp1_ABCD_. To further probe the mechanism by which Drp1_ABCD_ controls endocytosis, we examined the localization of the endocytic zone at the postsynaptic region using mCherry-clathrin light chain and Psd-95-Fibronectin intrabodies. We found that the loss of Drp1_ABCD_ dissociates mCherry-clathrin signals from Psd-95 signals in Drp1_ABCD_-KO neurons (Figure 3A and B). The new data suggest that Drp1_ABCD_ is not involved in creating the membrane curvature; rather, Drp1_ABCD_ helps to localize the endocytic zone next to the postsynaptic density (Figure 3C). Please see our related responses to reviewer 1 (Comment 1). In addition, we showed that the number of clathrin coated pits (CCPs) is increased in postsynaptic regions using electron microscopy (Figure 2). Taken together, these data suggest that clathrin may be partially depleted in the vicinity of the postsynaptic density, and therefore endocytosis is slowed in KO neurons (Figure 2S). Furthermore, electron microscopy showed that changes in the number of CCPs are only seen in postsynaptic regions but not presynaptic regions (Figure 2I, 2J, 2Q and 2R), suggesting that Drp1_ABCD_ is mainly involved in postsynaptic endocytosis.

Is this Drp1 variant recruited to the postsynaptic endocytic pit, or to the PM in general, in the absence of dynamin? Is there a role for Mff in the selective PM recruitment of this variant as alluded by Li et al., 2013? It is not clear what allows the selective recruitment of this Drp1 splice variant to the PM.

In our previous study, we showed that Drp1_ABCD_ is associated with the plasma membrane as well as late endosomes and lysosomes when expressed in Drp1-KO fibroblasts (in which total Drp1 is lost) (Itoh et al., 2018). During the revision process, to test whether the localization of Drp1_ABCD_ at the plasma membrane requires dynamin, we expressed Drp1_ABCD_ in Drp1-KO cells in the absence or presence of a dynamin inhibitor, dynasore. We found that the localization of Drp1_ABCD_ at the plasma membrane is similar in the presence or absence of dynasore (Figure S6A). Therefore, plasma membrane localization of Drp1_ABCD_ is likely independent of dynamin function. This is consistent with the notion that Drp1_ABCD_ controls the position of the endocytic zone likely upstream of dynamin function.

We also asked whether the plasma membrane localization of Drp1_ABCD_ requires Mff using Mff-KO fibroblasts. New results show that Drp1_ABCD_ is associated with the plasma membrane in Mff-KO cells similar to WT cells (Figure S6B). Li et al., 2013, showed that Drp1 is associated with Mff on synaptic vesicles at the presynaptic terminus. In the present study, we show that Drp1_ABCD_ functions at the postsynaptic terminus. Therefore, there may be multiple mechanisms that anchor different Drp1 isoforms to the membranes at different subcellular locations.

The recruitment and role of this variant should therefore be further investigated in the context of isoform-specific and pan-dynamin knockdown, which should provide insight into Drp1's role and mechanism, specifically in postsynaptic CME. What is the morphology of these endocytic pits in the absence (or presence) of dynamin 1/2/3? Can Drp1 substitute for dynamin in postsynaptic CME? Further, are there specific binding partners at the PM for this variant versus other Drp1 isoforms? In other words, what is the basis of the selective recruitment of this Drp1 splice variant over others?

As discussed above (Comment 1), our data suggest that the role of Drp1_ABCD_ is different from that of dynamin. Therefore, Drp1_ABCD_ would not replace dynamin in membrane constriction. We agree with this reviewer that molecules that recruit Drp1_ABCD_ to the plasma membrane are of great interest. We have ruled out the possibilities that Drp1_ABCD_ is recruited to the plasma membrane through dynamin or Mff (Comment 2). We plan to search for a plasma membrane receptor for Drp1_ABCD_ in our future studies and feel that the actual identification of such molecules will take a large body of new work and is beyond the scope of the present study. Along this line, we would prefer not to include the data about the role of dynamin and Mff in the plasma membrane localization of Drp1_ABCD_ in the current manuscript. Instead, we would like to present the data in our future study in which this mechanism is more specifically addressed with additional experiments. If this reviewer and editor feel that the data need to be included in the current study, we would be happy to do so. Please let us know.

In a nutshell, the following sentence from this manuscript requires additional clarification: "The accumulation of shallow and U-shaped CCPs, but not omega-shaped ones, suggest that Drp1_ABCD_ may function at an early step upstream of the constriction and severing of the neck of coated pits that is mediated by dynamin (Schmid and Frolov, 2011)."What is the role and mechanism of Drp1 here? This should be established to complete this manuscript, which will provide critical insights into the largely unstudied role of Drp1, outside of the context of mitochondria.

As discussed with reviewer 1 (Comment 1), we further analyzed CCPs using electron microscopy during the revision process. In the original manuscript, we showed that the number of CCPs is higher in KO neurons and that dynasore treatments did not increase their number (Figure 2M and N). Conversely, in WT neurons, the number of CCPs was increased when treated with dynasore because endocytosis was blocked at the last step by dynamin inhibition (Figure 2M and N). Therefore, it appears that Drp1_ABCD_ loss affects early steps during endocytosis (Figure 2S).

In the revised manuscript, we examined CCPs after NMDA stimulation. In the presence of dynasore, WT neurons increased the number of postsynaptic CCPs after NMDA stimulation, indicating that NMDA-promoted endocytosis was blocked at the step of membrane scission by dynamin (Figure 2Q). In contrast, KO neurons still did not increase the number of CCPs (Figure 2Q). These data further support the notion that NMDA-stimulated endocytosis is slower in Drp1_ABCD_-KO neurons.

Interestingly, when KO neurons were treated with NMDA in the absence of dynasore, the number of CCPs was decreased (Figure 2Q). In contrast, in WT neurons, the number of CCPs was slightly increased presumably as a result of a combination of increased initiation endocytosis and internalization of endocytic vesicles. These data suggest that, in KO neurons, NMDA still induces internalization of some endocytic vesicles, but early steps of endocytosis such as its initiation and progression remain slow. We think that KO neurons have decreased kinetics of endocytosis but do not completely block endocytosis (Figure 2S). This model is consistent with both the accumulation of shallow CCPs in KO neurons (Figure 2L). Decreased levels of clathrin in postsynaptic regions, as seen by clathrin and Psd-95 imaging (Figure 3A-C), may contribute to the slow kinetics. We have added this discussion to the revised manuscript.

Reviewer #3:The study by Itoh et al. investigates the function of the Drp1-_ABCD_ isoform, which contains all four alternative exons and is specific to the brain. The authors use both shRNA knockdown and CRISPR isoform specific knockouts. They show that Drp1-_ABCD_ is expressed during postnatal development, is enriched in dendritic spines, and regulates postsynaptic clathrin-mediated endocytosis. Drp1-_ABCD_ KO or KD induces primary neuron dendrites in culture and in utero KD in hippocampus increases primary neurites. Finally, Drp1-_ABCD_ KO mice display enhanced sensorimotor gating behavior in mice. This is a clear and well written study that provides novel information into Drp1-_ABCD_ function. However, additional studies would further support the findings and rule out a role for Drp1-_ABCD_ in mitochondria function.

We are grateful to this reviewer for the encouraging comments that our study provides novel information regarding Drp1_ABCD_ function. We also appreciate the constructive suggestions and concerns. During the revision process, we have addressed all of them as described below.

1) Please include representative images used to quantify Spine/Dendrite expression of HA-Drp1-_BCD_ (Figure 1E and F) similar to those shown of HA-Drp1-_ABCD_ in Figure 1G.

As recommended, we have included images for HA-Drp1_BCD_ in revised Figure 1H.

2) The mitochondria analysis in Figure 3 and S2 is preliminary. Since the main role of Drp1 is to mediate mitochondrial fission it will be important to determine if mitochondria are unaltered with Drp1-_ABCD_-KO and KD. The authors should provide quantification of mitochondria morphology, which may require genetic labeling of mitochondria. Additionally, Drp1 can alter mitochondria across dendritic regions. From the images it seems that only proximal dendrites are examined. It will be important to analyze the mitochondria across the dendritic field.

We agree with this reviewer that it is crucial to quantify mitochondrial morphology. In the revised manuscript, we included quantification of mitochondrial size in both proximal and distal regions along dendrites (Figure 3D-F). In addition, we measured mitochondrial size in the postsynaptic region using electron microscopy (Figure 2O and 2P. – dynasore). The new results from both immunofluorescence and electron microscopy show that the loss of Drp1_ABCD_ does not affect mitochondrial morphology.

3) In the last sentence of the Abstract the authors state that their results suggest that DRP1-_ABCD_ controls morphology and function through surface trafficking of neurotransmitter receptors at postsynapses. Inclusion of more details throughout the manuscript that relate the endocytosis findings to receptor trafficking would help support this conclusion and make it understandable to readers unfamiliar with the neurotransmitter receptor trafficking field.

Thank you for this important suggestion. As described in our responses to reviewer 1 (Comment 1) and reviewer 2 (Comment 1 and 4), we performed new experiments and substantiated our conclusion that Drp1_ABCD_ regulate postsynaptic endocytosis. However, we do not know what critical endocytic cargos are for dendritic morphogenesis at this moment. Therefore, we toned down the statements about the involvement of Drp1_ABCD_ in neurotransmitter receptor trafficking in the revised manuscript.

[Editors’ note: what now follows is the decision letter after the authors submitted for further consideration.]

We were helped in making our decision by the original reviewers, who read your revised manuscript. They all agreed that your changes had strengthened the story, and thought it would be helpful to include an illustration or model demonstrating how Drp1 _ABCD_ might function in postsynaptic endocytosis.

We are grateful to the reviewers for their supportive comments regarding our manuscript. As recommended, we have expanded a model for the function of Drp1_ABCD_ in the revised version of Figure 3G.

The reviewers also suggested a couple of additional experiments. First, there seems to be a discrepancy between the EM data in Figure 2Q, showing more CCPs at the postsynaptic terminal under basal conditions in the KO cells, and the light microscopy data in Figure 3 A-B, which appears to show less clathrin at the postsynaptic terminal terminal. The reviewer who pointed this out felt that you needed more direct evidence for your hypothesis that Drp1_ABCD_ regulates cargo recycling and suggested that you investigate the localisation of AMPA receptors in your KO cells.

We agree that this is an important point. Figures 3A and 3B show that a larger population of PSD95-marked synapses lack an endocytic zone in Drp1_ABCD_-KO neurons (30-35%) compared with WT neurons (15-20%). These numbers are similar to those reported for disrupted Homer, an adaptor protein that connect the postsynaptic density and endocytic zone (Lu et al., 2007). Our data suggest that Drp1_ABCD_ is important for the connection between postsynaptic density and the endocytic zone. However, the loss of Drp1_ABCD_ does not dissociate all of the endocytic zone from the postsynaptic density. Drp1_ABCD_ likely regulates the connection of postsynaptic density with a clathrin pool in a fraction of synapses (Figure 3G).

We speculate that decreased levels of clathrin in the synapses in Drp1_ABCD_-KO neurons would slow the progression of endocytosis. In these synapses, clathrin-coated pits initiate formation, but the maturation of clathrin-coated pits is likely decreased due to limited availability of clathrin molecules. As a result, coated pits are accumulated at relatively early stages of endocytosis (e.g., shallow and U-shaped clathrin-coated pits) (Figure 2L). This model could explain why the number of coated pits is increased while the levels of clathrin are decreased.

The suggested experiment to examine the localization of the AMPA receptor further helped us develop our model. In the revised manuscript, we analyzed the surface localization of GluR1, GluR2, and GluR3 using two different approaches. First, we labeled surface proteins using biotin in WT and Drp1_ABCD_-KO neurons and found that the surface levels of these receptors are similar in WT and KO neurons (Figure 3C and 3D). Second, we examined the internalization of GluR1 in response to chemical LTP using GluR1-GFP. We found that GluR1 was similarly internalized after chemical LTP in WT and KO neurons (Figure 3E and 3F). These data suggest that GluR1, GluR2, and GluR3 are not cargos for the endocytic pathway regulated by Drp1_ABCD_. We plan to identify cargos in the Drp1_ABCD_ pathway in our future study. We believe that the identification of such cargos is beyond the scope of the current study.

The other suggested experiment was to carry out a similar study to the one in supplementary Figure 6, showing a lack of effect of dynasore on mitochondrial morphology in MEFs, but this time using neurons as in Figure 3D. Because the labelled mitochondria often overlap, it will be important to show how you measure your mitochondrial length. Both of these experiments appear to be straightforward enough to complete within two months.

Thank you for suggesting another critical experiment. As suggested, we have analyzed mitochondrial morphology in hippocampal neurons after dynasore treatments. New data show that, consistent with the MEFs data, dynasore did not affect mitochondrial morphology in these neurons (Figure 2—figure supplement 1A-C). To minimize the effects of overlaps of mitochondria observed in 2D images, we used serial Z-sections to identify individual mitochondria in this analysis. We have added this description to the revised Materials and methods.

[Editors' note: further revisions were requested prior to acceptance, as described below.]

The manuscript has been improved but there are some remaining issues that need to be addressed before acceptance.Specifically, there is still the issue of the connection between the phenotypes, which has not been solved. The authors have convincing data regarding the effect of Drp1_ABCD_ deletion on post-synaptic endocytosis and the formation of ectopic dendrites. However, they do not show that ectopic dendrite formation occurs because postsynaptic endocytosis is affected. Therefore, the last sentence of the Abstract and the last paragraph of the Discussion must be changed.

As suggested, we have modified the last sentence of the Abstract and the last paragraph of the Discussion in the revised manuscript.